# AtmoSwing: Analog Technique Model for Statistical Weather forecastING and downscalING (v2.1.0)

Pascal Horton[1,2,3]

[1]University of Bern, Oeschger Centre for Climate Change Research, Institute of Geography, Bern, Switzerland
[2]University of Lausanne, Institute of Earth Sciences, Lausanne, Switzerland
[3]Terranum SARL, Bussigny, Switzerland

**Correspondence:** Pascal Horton (pascal.horton@giub.unibe.ch)

**Abstract.** Analog methods (AMs) use synoptic scale predictors to search in the past for similar days to a target day in order to infer the predictand of interest, such as daily precipitation. They can rely on outputs of numerical weather prediction (NWP) models in the context of operational forecasting or outputs of climate models in the context of climate impact studies. AMs require low computing capacity and have demonstrated a useful potential for application in several contexts.

5  AtmoSwing is an open source software written in C++ that implements AMs in a flexible way so that different variants can be handled dynamically. It comprises four tools: a Forecaster for use in operational forecasting, a Viewer to display the results, a Downscaler for climate studies, and an Optimizer to establish the relationship between predictands and predictors.

The Forecaster handles every required processing internally, such as NWP outputs downloading (when possible) and reading, grid interpolation, etc., without external scripts or file conversion. The processing of a forecast requires low computing efforts 10  and can even run on a Raspberry Pi computer. It provides valuable results, as revealed by a three-year-long operational forecast in the Swiss Alps.

The Viewer displays the forecasts in an interactive GIS environment with several levels of syntheses and details. This allows providing a quick overview of the potential critical situations in the upcoming days, as well as the possibility for the user to delve into the details of the forecasted predictand and criteria distributions.

15  The Downscaler allows using AMs in a climatic context, either for climate reconstruction or for climate change impact studies. When used for future climate studies, it is necessary to pay close attention to the selected predictors, so that they contain the climate change signal.

The Optimizer implements different optimization techniques, such as a semi-automatic sequential approach, Monte–Carlo simulations, and a global optimization technique using genetic algorithms. Establishing a statistical relationship between pre-20  dictors and predictands is computationally intensive because it requires numerous assessments over decades. To this end, the code was highly optimized for computing efficiency, is parallelized (using multiple threads) and scales well on a Linux cluster. This procedure is only required to establish the statistical relationship, which can then be used for forecasting or downscaling at a low computing cost.

# 1 Introduction

Approaches based on the concept of analogy are widespread in different domains of science and engineering. In hydrometeorology, it entails retrieving data on atmospheric conditions from the past that can be considered as similar to the situation at hand, with consequences that may be expected to be similar. The consequences can be local variables of interest such as the occurrence of fog, favourable conditions for avalanches, wind intensity, or the precipitation amount. The approach relies on the idea expressed by Lorenz (1956, 1969), that similar situations in terms of atmospheric circulation are likely to lead to similar local weather. AMs require at least two concurrent archives: one that provides the value of the local variable of interest called the predictand, and another one describing the past atmospheric situations through different variables called predictors, to which the situation at hand will be compared.

Usually, the predictand values could be derived by modelling the chain of processes linking the predictors to the predictand. The processes involved range from large-scale dynamical states of the atmosphere down to very small-scale microphysical processes. These require models that are extremely complex, data-intensive, and time-consuming. Conversely, given an appropriate set of predictor archives, a sufficient number of situations analogous to a target situation can be identified so that reasonable values can be obtained for the predictand, with low computing effort. This is particularly true for a specific predictand that is critical in hydrometeorological applications, namely, the precipitation amount over a given domain and time duration. The forecast provided by AMs is issued as a statistical distribution based on the observed predictand values from the selected analogs, unless only the single best analog is considered, which usually results in a lower skill (Bontron and Obled, 2005).

Analog methods (AMs) are used in two different types of approaches (Rummukainen, 1997): perfect prognosis, for which the statistical relationship is calibrated using observed predictors, and model output statistics (MOS), for which the relationship is calibrated using outputs of a specific climate or numerical weather prediction (NWP) model. AMs are often used to predict daily precipitation, either in an operational forecasting context (e.g. Guilbaud, 1997; Bontron and Obled, 2005; Hamill and Whitaker, 2006; Bliefernicht, 2010; Marty et al., 2012; Horton et al., 2012; Hamill et al., 2015; Ben Daoud et al., 2016) or a climate downscaling context (e.g. Zorita and von Storch, 1999; Wetterhall, 2005; Wetterhall et al., 2007; Matulla et al., 2007; Radanovics et al., 2013; Chardon et al., 2014; Dayon et al., 2015; Raynaud et al., 2016). Other predictands are also considered, such as precipitation radar images (Panziera et al., 2011; Foresti et al., 2015), temperature (Radinovic, 1975; Woodcock, 1980; Kruizinga and Murphy, 1983; Delle Monache et al., 2013; Caillouet et al., 2016; Raynaud et al., 2016), wind (Gordon, 1987; Delle Monache et al., 2013, 2011; Vanvyve et al., 2015; Alessandrini et al., 2015b; Junk et al., 2015b, a), solar radiation or power production (Alessandrini et al., 2015a; Bessa et al., 2015; Raynaud et al., 2016), snow avalanches (Obled and Good, 1980; Bolognesi, 1993), and the trajectory of tropical cyclones (Keenan and Woodcock, 1981; Sievers et al., 2000; Fraedrich et al., 2003). AMs are also used for seasonal forecast (Barnston et al., 1994; Xavier and Goswami, 2007; Charles et al., 2012; Wu et al., 2012; Shao and Li, 2013).

An AM was evaluated during the project STARDEX (*STAtistical and Regional dynamical Downscaling of EXtremes for European regions*, see Goodess, 2003; STARDEX, 2005). One of the goals of the project was to compare various downscaling

methods to determine weather extremes, and the AM was selected as being among the most useful techniques for daily precipitation (Maheras et al., 2005; Schmidli et al., 2007). Bliefernicht (2010) obtained superior results with AMs than downscaling methods based on weather typing.

The use of AMs for operational forecasting of daily precipitation originates in the work of Duband (1970, 1974, 1981). They were then designed for operational forecasting at EDF (Electricité de France) in order to better manage water resources and flood risks. They are used mainly by practitioners, notably hydropower companies (Desaint et al., 2008; Ben Daoud et al., 2009; Obled, 2014) or flood forecasting services in France and Switzerland (Marty, 2010; García Hernández et al., 2009; Horton et al., 2012). When comparing the results from AMs to an ensemble forecast, Marty (2010) found AMs to be better than the considered ensemble, particularly for strong precipitation. However, AMs should not be considered as a substitute for NWP models, but as a complement in order to obtain a fast statistical adaptation that is known to be accurate several days in advance. Therefore, they contribute to the analysis of potentially critical situations in flood forecasting, for example, and are very useful in early warning.

Hamill and Whitaker (2006) used an analogy-based approach on the GFS reforecasts in order to correct systematic errors in the ensemble forecasts of temperature and precipitation. These biases could be corrected by taking into account the intrinsic local climatology provided by the AM. Moreover, the under-dispersion of the ensemble forecast from the numerical model has also been corrected using analogs (Hamill and Whitaker, 2006). Correction of ensemble forecast under-dispersion using AMs is also used operationally at EDF (Électricité de France).

The present work does not introduce a new method, but a software named AtmoSwing that implements AMs in a versatile and efficient way. It is versatile in that it facilitates the building of AM structures in a dynamic way, and because the code is written with an object-oriented architecture. It is efficient because it is written in C++ and leverages parallel computing. AtmoSwing is made up of different modules targeted either for operational forecasting (the Forecaster and the Viewer) or for climate impact studies (the Downscaler). Additionally, a module (the Optimizer) is available for calibrating the different parameters of the method. AtmoSwing is continuously evolving and has been used in Horton et al. (2012, 2017a, b, 2018) and Horton and Brönnimann (2018).

Some existing AMs designed for daily precipitation will first be described along with the required data (Sect. 2) and the software will then be presented (Sect. 3) together with the details of the modules: the Forecaster (Sect. 3.3), the Viewer (Sect. 3.4), the Downscaler (Sect. 3.5), and the Optimizer (Sect. 3.6). Section 4 discusses the parameters space of AMs through different calibration techniques and Sect. 5 provides a feedback from operational precipitation forecasting in the Swiss Alps. Some limitations of the AM are discussed in Sect. 6. The conclusions (Sect. 7) provide some additional perspectives for future developments of AtmoSwing.

## 2 Data and methods

### 2.1 Required data

AMs generally require three datasets: the historical predictand values, the historical predictor values for the same period and the predictors describing the target situation.

The predictand is often a daily or 6-hrly time series. One of the most used predictand is the daily precipitation, which is usually averaged over subregions in order to smooth local effects (Obled et al., 2002; Marty et al., 2012). These time series can be normalized by the precipitation value for a certain return period (for example 10 years, Djerboua, 2001) to allow for an easier comparison between subregions subject to different precipitation regimes.

In the early days of AMs in operational forecasting, the predictors were based on radio sounding data. Nowadays, the predictors' archive is often a global atmospheric reanalysis dataset, which provides gridded large-scale variables at any location in the world. Reanalyses are produced using a single version of a data assimilation system coupled with a forecast model constrained to follow observations over a long period. They provide multivariate outputs that are physically consistent, which contain information on the locations where few or no observations are available, including variables that are not directly observed (Gelaro et al., 2017). Even though reanalyses are considered as very accurate in a data-rich region such as Europe, they can have a non-negligible impact on the skill of the AMs, that can be even higher than the choice of the predictor variables (Dayon et al., 2015; Horton and Brönnimann, 2018). AtmoSwing can read eleven different reanalyses (Table 1), and others can be easily added thanks to the encapsulation of the dataset characteristics in the objects. Recommendations for the selection of a reanalysis can be found in Horton and Brönnimann (2018). Other predictor archives can also be used such as Sea Surface Temperature (SST, Reynolds et al., 2007). Bontron (2004) proposed that the minimum length of the archive should be 30 years for the prediction of daily precipitation under usual conditions, and 40 years or more for heavy rainfall. For smaller time-steps, shorter archives can be used (Horton et al., 2017b).

The predictors' dataset that describes the target situation varies according to the application of the AM. For operational forecasting (Sect. 3.3) they are outputs of NWP models such as the European Centre for Medium–Range Weather Forecasts (ECMWF) Integrated Forecasting System (IFS) or the National Centers for Environmental Prediction's (NCEP) Global Forecast System (GFS, Kanamitsu et al., 1991; Kanamitsu, 1989). For climate impact studies (Sect. 3.5), they are outputs of general circulation models (GCMs) or regional climate models (RCMs), such as the Coupled Model Intercomparison Project Phase 5 (CMIP5, Taylor et al., 2012) and EURO-CORDEX (Jacob et al., 2014).

### 2.2 Analog methods for daily precipitation

AtmoSwing does not rely on a single structure of the AM, but can implement different variants. A non-exhaustive selection of methods developed for different regions will be presented hereafter, focusing on the prediction of daily precipitation.

### 2.2.1 Characteristics of the AM

*Definition of the analogy* – The AM is based on the principle that two similar synoptic situations may produce similar local effects (Lorenz, 1956, 1969). The perfect analogy does not exist, but sufficiently similar situations leading to similar effects can be identified. To be relevant, this analogy must be selected by optimizing the following elements:

- The meteorological variables (predictors) must contain synoptic scale information with a direct or indirect dependency with the target predictand.

- The pressure (or isentropic) levels at which the predictors are selected.

- The spatial windows are the domains over which predictors are compared.

- The temporal windows are the hours of the day at which the predictors are considered when the time step of the predictors is smaller than the one of the predictand.

- The analogy criteria are distance measures used to rank past situations according to their degree of similarity with the target situation.

- Possible weights between the predictors (e.g., Horton et al., 2017b; Junk et al., 2015b).

- The number of analog situations $N_i$ to retain for the analogy level $i$.

*Seasonal preselection* – Lorenz (1969) restricted the search for analog situations to the same period of the year to cope with seasonal effects. This preselection is now often implemented as a moving selection of $\pm 60$ days centred around the target date, for every year of the archive (Table 2, Bontron, 2004; Marty et al., 2012; Horton et al., 2012; Ben Daoud et al., 2016). Alternatively, the candidate dates can be selected based on similar air temperature at the nearest grid point (Table 2, Ben Daoud et al., 2016).

*Analogy of atmospheric circulation* – A conditioning by variables describing the atmospheric circulation is present in a vast majority of AMs. The geopotential field (Z) is often used as a predictor since Lorenz (1969), who based the analogy on the levels 200, 500, and 850 hPa. Several pressure levels were later assessed by means of various criteria for the analogy based on the geopotential field (Duband, 1970, 1974, 1981; Guilbaud, 1997). It was found to be important to calculate the analogy for multiple pressure levels and different temporal windows (reference time of the predictors as they are usually available at a 6-hrly temporal resolution or higher) instead of a unique selection (Guilbaud and Obled, 1998; Obled et al., 2002). Bontron (2004) showed that the choice of the temporal window can be more important than the choice of the atmospheric level for daily precipitation (usually measured between 6 h UTC and 6 h UTC the following day). He concluded that the coupled geopotential heights at 1000 hPa (Z1000) at 12 h UTC and 500 hPa (Z500) at 24 h UTC provided the best performance (for a subset of the NCEP/NCAR Reanalysis I – NR-1; Kalnay et al., 1996; Kistler et al., 2001) for the investigated regions in France (Table 2). The analogy of the atmospheric circulation proposed by Bontron (2004) is still used operationally at the time of writing. Marty (2010) tested other temporal windows for intraday application on the basis of a more comprehensive reanalysis dataset

and proposed to change the hours of observation to 06 h UTC and 18 h UTC. Horton et al. (2018) showed that a selection of four combinations of pressure levels and temporal windows instead of two for the geopotential height improves the skill of the method (4Z, Table 2). The pressure levels and temporal windows were automatically selected by genetic algorithms for the upper Rhône catchment in Switzerland.

*Additional levels of analogy* – Additional levels of analogy are subsequent steps that subsample a lower number of analog situations from the antecedent level of analogy, based on other variables. A second level of analogy was first introduced by Mandon (1985) and Vallée (1986) based on wind, moisture variables, or temperature. Gibergans-Báguena and Llasat (2007) used the same kind of variables along with stability indexes. After a systematic assessment of the variables provided by NR-1, Bontron (2004) noted that a moisture index (MI) based on the product of the relative humidity at 850 hPa (RH850) and the total

precipitable water (TPW) provided the best skill (Table 2). Marty (2010) selected the MI at 925 hPa instead of 850 hPa and also considered the moisture flux (MF) at 700 or 925 hPa (Table 2). The MF is the product of the MI with the wind intensity. Horton et al. (2018) determined that the MI at 600 and 700 hPa were more useful than MF after the circulation analogy was applied to the four atmospheric levels (Table 2). Ben Daoud et al. (2016) also reconsidered the parameters of the MI and ended up with both 925 hPa and 700 hPa levels (Table 2). Subsequently, they added an additional level of analogy between the circulation and

the moisture analogy (Table 2) based on the vertical velocity at 850 hPa (W850). This AM, termed "SANDHY" for Stepwise Analogue Downscaling method for Hydrology (Ben Daoud et al., 2016; Caillouet et al., 2016), was primarily developed for large and relatively flat/lowland catchments in France (Saône, Seine).

    *Analogy criteria* – In early applications of AMs, the geopotential height was condensed using principal component analysis (PCA) and the selection of analog situations was performed according to a Euclidean distance in the space of the PCA. Guilbaud

(1997) stopped using PCA to work directly with the raw data interpolated on grids, which resulted in an improvement. In the case of variables that describe atmospheric circulation, the Teweles–Wobus (S1) criterion (Eq. (1), Teweles and Wobus, 1954; Drosdowsky and Zhang, 2003) was identified as the most suited criteria based on different studies (Wilson and Yacowar, 1980; Woodcock, 1980; Guilbaud and Obled, 1998; Bontron, 2004). S1 allows for a comparison of the gradients and thus an analogy of the atmospheric circulation instead of considering the actual values at the grid points. For other predictors, classic criteria

representing Euclidean distances between grid point values are used: Mean Absolute Error (MAE) and Root Mean Squared Error (RMSE), the latter being used most often.

$$S1 = 100 \frac{\sum_i |\Delta \hat{z}_i - \Delta z_i|}{\sum_i max\{|\Delta \hat{z}_i|, |\Delta z_i|\}} \tag{1}$$

where $\Delta \hat{z}_i$ is the gradient between the *i*th pair of adjacent points from the geopotential field of the forecasted target situation, and $\Delta z_i$ is the corresponding observed geopotential gradient in the candidate situation. The differences are processed separately

in both directions. The smaller the S1 values, the more similar the pressure fields. AtmoSwing allows processing real gradients by taking into account the actual distance between points, or simple height differences by ignoring the horizontal distance.

Under the latitudes of central Europe, the impact of neglecting the horizontal distance is small (not shown), but it can become more important at higher latitudes.

*Other parameters* – The predictors are compared on a defined spatial window, which must be optimized to maximize the useful information and minimize noise. The spatial window is usually considered unique for all predictors of a level of analogy. Using genetic algorithms, Horton et al. (2018) introduced different spatial windows between the pressure levels, which increased the skill. Additionally, a weighting between the predictors was also successfully added instead of a simple equal-weights averaging. The number of analogs to select at each level of analogy should be optimized to be the best trade-off between taking into account local variability and maximizing useful synoptic information. It depends on the predictor dataset, the size of the spatial window and the length of the archive (Ruosteenoja, 1988; Van Den Dool, 1994).

*Probabilistic forecast* – After the last level of analogy, the observed values of the predictand of interest (here daily precipitation amounts) of the $N_i$ resulting dates provide the empirical conditional distribution considered as the probabilistic forecast for the target day. The empirical frequencies are processed for every predictand value after classification, based on the Gringorten parameters (for a Gumbel or exponential law; see Gringorten, 1963) and a probabilistic model can eventually be fitted (e.g. Gamma function, Obled et al., 2002). The forecast is finally often synthesized according to percentiles 20, 60 and 90 % (Guilbaud, 1997; Guilbaud and Obled, 1998).

*Use in operational forecasting* – In one of the very first uses in operational forecasting, radiosonde observations were used as predictors to predict precipitation for the next two days. However, because of the chaotic nature of the atmosphere, two analog situations quickly diverge over time (Lorenz, 1969). Thus, the AM has strong limitations regarding the analogy of temporal trajectories (Bontron, 2004). Given the superior capability of numerical models for simulating the dynamic evolution of the atmosphere, their outputs are now used as predictors for the coming days. The search for analogy thus aims to connect the forecasted synoptic situation with a local predictand, especially precipitation, which is more difficult to simulate for numerical models. When using AMs in operational forecasting, it should be noted that some variables such as moisture or vertical velocity might not be accurately predicted after a lead time of a few days due to higher uncertainties. Predictors describing the atmospheric circulation are generally considered to be more reliable.

## 2.2.2 Regional characteristics

The optimal predictors vary from one region to another, along with the leading atmospheric processes. Thus, the method needs to be adapted to the local conditions, available data, and to the size of the region of interest. Even for two locations that are close to each other but subject to different critical atmospheric conditions, the selection of the best predictors can vary. This is illustrated in Fig. 1 for two subregions of the Rhône catchment in Switzerland. For both subregions, all variables of NR-1 were assessed by optimizing the spatial window and the number of analogs for each one of them using the sequential calibration tool implemented in AtmoSwing (Sect. 3.6.4). The main similarities in the selection of the best predictor from NR-1 at both locations are: (1) the variables describing the atmospheric circulation (pressure fields or geopotential heights) perform best, and (2) they are better when compared with the S1 criteria (asterisk in Fig. 1) instead of the RMSE. The main difference is that the pressure fields better explain the precipitation when they are considered close to the ground for the Chablais region, and at

a higher altitude for the South-east crests. This is driven by the elevation of the stations and by the main atmospheric drivers related to the precipitation at these locations.

The choice of the best predictors is likely to vary from one reanalysis dataset to another. This comprehensive comparison was not repeated with other datasets, because a selection of the best predictors using genetic algorithms would be less cumbersome
(Sect. 3.6.5).

### 2.2.3  Method nomenclature

Variants of the AMs are numerous and it is not always easy to reference them in a short and descriptive way. In AtmoSwing, a basic nomenclature is used (Fig. 2) in order to express the structure into a simple identifier. This cannot describe all the parameters of the AM, but quickly illustrates the structure of the method. This is particularly useful when working with a
global optimization method, where nothing is fixed but the structure of the AM. This nomenclature has been used in Horton et al. (2017a, b, 2018) and Horton and Brönnimann (2018).

The naming contains different blocs (separated by a hyphen) for the various levels of analogy. It starts with the specification of the preselection (P; can be omitted when comparing AMs with the same preselection approaches), which can be one of two types:

– PC: calendar period (±60 days around the target date)

– PT: based on air temperature (Ben Daoud, 2010)

Then, the following levels of analogy are listed, which may start with an optional A (for analogy). For every level of analogy, the number of variables used (combination of atmospheric levels and time of observation) is first provided, and then the short name of the variable is given (according e.g. to ECMWF conventions; in upper case), for example:

– Z: geopotential (circulation)

– TPW: total precipitable water

– RH: relative humidity

– V: wind velocity

– W: vertical velocity

– MI: moisture index (TPW * RH)

– MF: moisture flux (V * TPW * RH)

In order to keep the identifier simple, no value of atmospheric level or time of observation is specified. Moreover, the analogy criterion is not specified and is supposed to be S1 for Z and RMSE for the other variables. If anything changes from these conventions, it can be noted as a flag. The flag (lower case) can also provide other information, such as the optimization
method:

- sc : sequential calibration (can be omitted as considered as default, see Sect. 3.6)

- go (or just "o"): global optimization (by means of genetic algorithms for example)

This nomenclature can be adapted to specific needs or simplified for better readability (e.g. by removing the specification of the preselection). Examples can be found in Table 2.

## 5    3    AtmoSwing

AtmoSwing is made of 4 main modules that are standalone, but do share a common code basis: the Forecaster for operational forecasting, the Viewer for displaying the forecast in a GIS environment, the Downscaler for climate applications, and the Optimizer that is used to establish the statistical relationship that defines the analogy for a given predictand. Separating the Forecaster and the Viewer allows for automation of the forecast on a server and the local display of the results. The Forecaster,
the Downscaler and the Optimiser can be used either with a graphical user interface or a command-line interface.

### 3.1    Technical aspects

The code is written in object-oriented C++ and relies on the wxWidgets (Smart et al., 2006) library to provide a cross-platform native experience to users. CMake is used to build AtmoSwing under Windows, Linux, or Mac OSX. Developments have been partly performed using a test-driven development (TDD) approach. Continuous integration has been set up (on Travis
CI and AppVeyor) so that a collection of more than 600 tests can be evaluated on the three operating systems every time new code is pushed to the server, to prevent regressions. Every analogy criterion, performance score, searching and sorting functions, data manipulation, etc., are tested. Some tests specific to the AM rely on the results of another analog sorting software developed at the Université Grenoble Alpes. They ensure that the results of AtmoSwing are exactly equivalent to this model, given the same parameters and data. The source code is under version control (Git) and is open source (on GitHub,
www.atmoswing.org, Horton, 2018a). The GitHub organization page (https://github.com/atmoswing) also contains toolboxes to work with the outputs of AtmoSwing in R (Horton and Burkart, 2018) or Python (Horton, 2018b).

Although processing an analog adaptation for a given target date is fast, numerous hindcasts over periods of several decades must be performed for calibration, which may become very time-consuming. Thus, great effort has been focused on minimizing the processing time using profiling tools. Firstly, all identified redundancies in the processing were removed. Then, when
searching for a certain date or data, the search starts in the region where it is likely to be found instead of exploring an entire array. Similar data are not loaded twice, but instead shared pointers are used. Several other improvements allow reducing the computing time, for example the use of the quicksort method (Hoare, 1962) to sort the date vectors according to the analogy criterion. Different implementation variants were tested in order to select the most efficient approach: for example, when storing analog dates according to their criterion value, it is faster to insert them in a fixed-size array instead of storing them all and
subsequently sorting the array. When using the S1 criteria, the gradients are pre-processed on the predictor data, so that they are only processed once. AtmoSwing also uses the linear algebra library Eigen 3 (Guennebaud et al., 2010) for calculations on

vectors and matrices, which result in time-saving. Multi-threading is also implemented so that the search for analog situations in the archive is distributed among the available threads.

A user interface allows for the creation of the predictand database in the NetCDF format from text files. During the process, Gumbel adjustments are automatically calculated for precipitation data to determine the values corresponding to different
return periods. The time series are normalized using a selected return period (default 10 years) and their square root can be processed. The final database file contains both the raw and the normalized series, as well as characteristics of the gauging stations and some metadata.

## 3.2    Modular approach and implementation

AtmoSwing's great strength is that it is designed to process the analog method in a modular fashion. The structure of the AM
(number of analogy levels, number of predictors) is built dynamically (Fig. 3), and nothing is fixed a priori. The software then successively performs as many analogy levels as the user specifies, using all the predictors indicated. Each level of analogy results in an object containing target dates, analog dates, values of the analogy criteria, values of the predictand (at the final stage), and other data. This object can be saved as a NetCDF file and/or can be injected into a new analogy level. The whole structure of the AM is defined through an XML file. Even the time step of the method (6 or 24 hours for example) is a dynamic
parameter.

Each implementation of the AM (see Sect. 2.2) may enter this scheme, even if it consists of pre-processed variables (e.g. moisture index). Various pre-processing functions are implemented as the calculation of the moisture index or flux, multiplication operations, or calculation of the gradients. The user can dynamically specify the pre-processing method and the predictors to use in the XML file.
This modular approach is implemented through object-oriented programming, as a direct consequence of polymorphism. This allows, for example, processing of a predictor object as a single interface to entities representing any reanalysis dataset. Similarly, the criterion can be of different types, as well as the score for calibrating. The different types of objects that are instantiated are defined in the XML parameters file. Thus, there is a single implementation of the analog method capable of interacting with different types of objects in various contexts (calibration, forecasting, downscaling).

## 25    3.3    AtmoSwing Forecaster

The Forecaster module allows processing of operational forecasts. The software can be compiled with a graphical user interface (GUI), or without it to be used on a headless server through a command line interface (CLI). Processing a forecast requires very low computing capabilities and can be performed on a low-end computer. It successfully runs on a Raspberry Pi 3 (Model B).
To this day, the software can use the outputs of IFS or GFS (see Sect. 2.1). When possible, it first downloads the relevant model outputs and interpolates the gridded data to match the resolution of the archive. The analogs dates are next extracted according to the selected AM variant and the predictand data are associated with the corresponding dates. The results are finally saved in auto-describing NetCDF files. If requested, a synthetic XML file is generated for easier integration on a web platform,

for example. Every step of the forecast, from predictor downloading (when possible) to the final results, is performed in the software (and controlled through configuration), without the use of external scripts (e.g. for data conversion).

Both the GUI and the CLI facilitate the processing of a forecast based on the most recent NWP outputs, or for a given date or period. When there is no new predictor data available, the forecast is not processed and computing resources are not consumed. The recommended use is thus to set up an automatic task on a server to trigger the forecast every 30 minutes. This would for example provide four forecasts a day.

Before being used in operational forecasting, the AMs were calibrated in a perfect prognosis framework, usually using a reanalysis dataset (Sect. 3.6). However, this does not take into account the uncertainty related to the forecast of the target situation by NWP models. One might be willing to take into account this uncertainty, which increases with the lead time. A solution is to increase the number of analog situations with the lead time, which should be optimized for every lead time on a forecast archive or a reforecast dataset (Thevenot, 2004). This technique is available in AtmoSwing, as the number of analogs can be specified for every lead time.

A meteorological variable that was proven as a good predictor in the perfect prognosis framework may eventually be poorly predicted by the selected NWP beyond a certain lead time. It should then be dropped after this lead time. For example, when using moisture variables for the second level of analogy, Thevenot (2004) showed that beyond a lead time of three days the AM with two levels did not perform better than the one with a single level of analogy. Datasets of reforecast from the selected NWP models allow assessing these aspects for different lead times.

## 3.4 AtmoSwing Viewer

AtmoSwing Viewer allows for the display of the files produced by the Forecaster in an interactive GIS environment (Fig. 4) with several levels of synthesis. It first provides an overview of possible alerts using colour codes on the lead time switcher (upper right in the GUI, see Fig. 4) that represent the worst case scenarios, or in the alarm panel (on the left side of the GUI). The alarm panel allows for a synthesis of the highest forecasted values for the different AMs and the different lead times. By default, the colours are expressed relative to the 10 year return period, for the $90^{th}$ percentile (which can be changed in the preferences). This highest level of synthesis allows for quick identification of potentially critical situations in the days ahead.

Then, the user can explore the forecasts in more details, starting from the provided map (Fig. 4). The map displays the forecast of the selected AM variant (selected in the upper left panel) and the selected lead time (upper right). During the forecast, one AM might have parameters that differ by subregions, such as the number of analogs or the spatial windows. The Viewer automatically gathers the similar AM types and provides a composite view of the optimal forecasts per subregion. The user can, however, choose to display the results associated with a single parameter set for the entire region (by opening the tree view and selecting a child element), which provides a homogeneous set of analog dates. A display of all lead times on a single map is possible based on a symbolic representation on a circular band with a box for every lead time (Fig. 5). The number of boxes is adjusted to the number of lead times. This representation offers a global spatiotemporal visualization for a chosen AM.

Colour scales in the map can be adjusted by choosing (on the left part of the GUI) the predictand reference (raw value or ratio to different return periods) and the quantile of the distribution. Using a ratio to a certain return period eases the interpretation of the expected precipitation given that reference values can drastically differ from one location to another, particularly in mountainous regions. All information relative to a rain gauge station (or catchment), such as its location, its name, or the values of different return periods, are stored in the forecast files to be displayed for end users who do not have the predictand database.

By clicking on a station on the map (or by selection from a dropdown list on the left), a new window appears with a plot of the forecasted time series (Fig. 6). By default, the plot contains the usual three considered percentiles ($90^{th}$, $60^{th}$, and $20^{th}$), along with the 10 best analogs (crosses) with a colour code from yellow (tenth) to red (first). The 10-year return period value is also displayed as a red line. The user can choose to hide any data or to display supplementary information (all analogs, all $10^{th}$ percentiles, or all return periods) in the left panel. Traces of previous forecasts are also automatically loaded and displayed to provide information on the consistency of the forecasts.

The user can then delve into further details and display the predictand cumulative distribution for a given lead time (Fig. 7). This can inform if there is a shift between the distribution of all analogs versus the 10 best. Such a shift warns of a risk of under/overestimation when considering the full distribution, particularly for high precipitation amounts. Indeed, the number of extreme precipitation events in the archive is limited and they are thus likely to be underrepresented in the selected analog dates. Different authors have shown that if the $60^{th}$ percentile is best to forecast the occurrence and the amount of precipitation for common situations, the $90^{th}$ percentile is a better indicator for strong to extreme events (Djerboua, 2001; Bontron, 2004; Marty, 2010). It is, therefore, necessary to pay close attention when the $90^{th}$ percentile reaches high values, as this may be indicative of possible extreme precipitations due the presence of several analog dates with high precipitation amounts in the distribution (Djerboua, 2001).

The distribution of the analogy criteria (not shown) can also be displayed to identify eventual discontinuities in the criteria values. Finally, one can display the analog dates with the corresponding predictand and criteria values in an interactive spreadsheet (not shown).

AtmoSwing Viewer relies on workspaces defined in XML files to specify the path to the forecast directories and the GIS layers. It is thus easy to switch from a forecast for a region to another. Many GIS formats are supported thanks to GDAL (Geospatial Data Abstraction Library, GDAL Development Team, 2014). A user can have as many layers as desired and can control their display properties (colour, transparency).

### 3.5 AtmoSwing Downscaler

The Downscaler module is the last addition to AtmoSwing. Its purpose is to downscale either climate model outputs for climate impact studies or reanalyses for climate reconstruction of the past.

The Downscaler is able to read outputs of general circulation models (GCMs) or regional climate models (RCMs), such as the Coupled Model Intercomparison Project Phase 5 (CMIP5, Taylor et al., 2012) and EURO-CORDEX (Jacob et al., 2014), and can be extended to other datasets. CMIP5 and EURO-CORDEX are distributed in the NetCDF format, but present a great

variety of time steps, temporal references, spatial resolution, and file structures. A complete redesign of the management of the predictor data was necessary to provide the flexibility required to account for this variety. The Downscaler is thus able to parse these datasets original files by exploiting the self-descriptive capacity of NetCDF files.

The use of AMs in the context of future climate is rather new. Not all AMs can be used for this purpose, because some predictors might not capture the climate change signal well and the preservation of the relationship between predictors and predictands must prevail. However, several authors have demonstrated the transferability of some AMs for future climate (Dayon et al., 2015, 2018; Raynaud, 2016; Turco et al., 2017). The transferability of an AM must be assessed before it is used in such a context.

AMs have also been used to perform climate reconstruction of the past (Caillouet et al., 2016, 2017; Bonnet et al., 2017). Such applications allow, for example, hydrological modelling of flood events in periods where no meteorological data are available, or analysis of past severe droughts.

## 3.6 AtmoSwing Optimizer

AtmoSwing Optimizer is a single tool that integrates different optimization methods, presented in Sect. 3.6.3 to 3.6.5. Its purpose is to establish the statistical relationship between the predictors and a predictand. The calibration framework is detailed in Sect. 3.6.1 and the implemented skill scores are listed in Sect. 3.6.2.

### 3.6.1 Calibration framework

The calibration of the AM is usually performed in a perfect prognosis (Klein et al., 1959) framework (Bontron, 2004; Ben Daoud, 2010). Perfect prognosis uses observed or reanalyzed data to calibrate the relationship between predictors and predictands, as opposed to the MOS approach that establishes the relationship based on model outputs. As a result, perfect prognosis yields relationships that are as close as possible to the natural links between predictors and predictands. However, there are no perfect models and even reanalysis data may contain biases that cannot be ignored (Dayon et al., 2015; Horton and Brönnimann, 2018). Thus, the considered predictors should be as robust as possible, i.e., they should have minimal dependency on the model. With MOS approaches, reforecasts can be used to establish the relationship between predictors and predictands, provided that the archive is long enough. However, the calibration procedure must be performed every time a new version is available in order to reduce the bias (Wilson and Vallée, 2002).

A statistical relationship is established with a trial and error approach by processing a forecast for every day of a calibration period. A certain number of days close to the target date are excluded to consider only independent candidate days. Validating the parameters of AMs on an independent validation period is very important to avoid over-parametrization and to ensure that the statistical relationship is valid for another period. In order to account for climate change and the evolution of measuring techniques, it is recommended that a noncontinuous period for validation should be used, distributed over the entire archive (Ben Daoud, 2010; Horton and Brönnimann, 2018). AtmoSwing's users can thus specify a list of the years to set apart for the validation that are removed from possible candidate situations. At the end of the optimization, the validation score is processed automatically.

### 3.6.2 Implemented performance scores

Multiple scores are implemented in AtmoSwing Optimizer and are listed hereafter. Details are only provided for the CRPS (Continuous Ranked Probability Score, Brown, 1974; Matheson and Winkler, 1976; Hersbach, 2000), which is most often used.

*Discrete deterministic predictions* - These are, for example, deterministic predictions of threshold exceedances. The continuous probabilistic nature of an ensemble of analogs can be transformed into a discrete prediction by considering a fixed percentile from the distribution, which is compared to a threshold exceedance of the predictand. On the basis of a contingency table (Wilks, 2006), multiple scores can be processed with AtmoSwing:

- Proportion correct (Finley, 1884)
- Threat Score (Gilbert, 1884)
- Bias
- False Alarm Ratio
- Hit Rate or Probability of Detection
- False Alarm Rate
- Heidke Skill Score (Heidke, 1926)
- Peirce Skill Score (Peirce, 1884)
- Gilbert Skill Score or Equitable Threat Score (Gilbert, 1884)

*Continuous deterministic predictions* - These types of predictions must be evaluated using distance measures. For AMs, the provided distribution is summarized by a chosen percentile, which is compared to the predictand value. Available scores are as follows:

- Mean Absolute Error
- Root Mean Squared Error

*Discrete probabilistic predictions* – Here, the probability of occurrence or the probability of belonging to a certain category is considered. The implemented scores are as follows:

- Brier Score (Brier, 1950)
- ROC diagram (Relative Operating Characteristic or Receiver Operating Characteristic, Mason, 1982)
- RPS (Ranked Probability Score, Epstein, 1969)

– SEEPS (Stable Equitable Error in Probability Space, Rodwell et al., 2010, 2011)

*Continuous probabilistic predictions* - These types of predictions are issued in the form of the expected statistical distribution for a variable, which needs to be compared to an observed value. This is the situation encountered when using multiple analogs from AMs.

Most assessment of AMs performance use the CRPS (Continuous Ranked Probability Score, Brown, 1974; Matheson and Winkler, 1976; Hersbach, 2000). It allows for evaluation of the predicted cumulative distribution functions $F(y)$, for example, the precipitation values $y$ associated with the analog situations, compared to the single observed value $y^0$ for a day $i$:

$$CRPS_i = \int\limits_0^{+\infty} \left[ F_i(y) - H(y - y_i^0) \right]^2 dy \tag{2}$$

where $H(y - y_i^0)$ is the Heaviside function that is null when $y - y_i^0 < 0$, and has the value 1 otherwise; the better the prediction, the lower the score. This score is now commonly used for the evaluation of continuous variable prediction systems (Casati et al., 2008; Marty, 2010). It can be decomposed into several indicators also implemented into AtmoSwing Optimizer, such as: reliability – resolution / uncertainty (Hersbach, 2000), or sharpness – accuracy (Bontron, 2004).

Its skill score expression is often used, with the climatological distribution of precipitation as the reference. The CRPSS (*Continuous Ranked Probability Skill Score*) is thus defined as follows (Bradley and Schwartz, 2011):

$$CRPSS = 1 - \frac{\overline{CRPS}}{CRPS_{clim}} \tag{3}$$

where $CRPS_{clim}$ is the CRPS value for the climatological distribution. A better prediction is characterized by an increase in CRPSS.

Finally, the rank diagram (Talagrand et al., 1997) and its accuracy as defined by Candille and Talagrand (2005) are also available.

### 3.6.3 The sequential calibration

The calibration procedure that we call "sequential" or "classic" was elaborated upon by Bontron (2004) (see also Radanovics et al., 2013; Ben Daoud et al., 2016). It is a semi-automatic procedure that optimizes the spatial windows in which the predictors are compared and the number of analogs for every level of analogy. The different analogy levels (e.g. the atmospheric circulation or moisture index) are calibrated sequentially. The procedure consists of the following steps (Bontron, 2004):

1. Manual selection of the following parameters:

    (a) Meteorological variable

    (b) Pressure level

     (c) Temporal window (hour of the day)

     (d) Number of analogs

2. For every level of analogy:

     (a) Identification of the most skilled unitary cell (four points for the geopotential height when using the S1 criteria and one point otherwise) of the predictor data over a large domain. Every point or cell of the full domain is assessed based on the predictors of the current level of analogy.

     (b) From this most skilled cell, the spatial window is expanded by successive iterations in the direction of the largest performance gain until no further improvement is possible.

     (c) The number of analog situations $N_i$, which was initially set to an arbitrary value, is then reconsidered and optimized for the current level of analogy.

3. A new level of analogy can then be added based on other variables such as the moisture index at chosen pressure levels and hours of the day. The procedure starts again from step 2 (calibration of the spatial window and the number of analogs) for the new level. The parameters calibrated for the previous analogy levels are fixed and do not change.

4. Finally, the numbers of analogs for the different levels of analogy are reassessed. This is performed iteratively by varying the number of analogs of each level in a systematic manner.

The calibration is performed in successive steps for a limited number of parameters with the aim of minimizing error functions or maximizing skill scores. Except for the number of analogs, previously calibrated parameters are generally not reassessed. The benefit of this method is that it is relatively fast, it provides acceptable results, and it has low computing requirements.

Small improvements were incorporated into this method in AtmoSwing Optimizer, then termed as "classic+", by allowing the spatial windows to perform other moves, such as: (1) increase in 2 simultaneous directions, (2) decrease in 1 or 2 simultaneous directions, (3) expansion or contraction (in every direction), (4) shift of the window (without resizing) in 8 directions (including diagonals), and finally (5) all the moves described above, but with a factor of 2, 3, or more. For example, an increase by two grid points in one (or more) direction is assessed. This allows skipping one size that may not be optimal. These supplementary steps often result in spatial windows that are slightly different. The performance gain is rather marginal for reanalyses with a low resolution such as NR-1, but might be more consistent for reanalyses with higher resolutions due to the presence of more local minima.

### 3.6.4 Variables exploration

The sequential calibration can also be used to explore the variables of a dataset. A list of variables, pressure levels, and temporal windows can be provided and all combinations are assessed through the classic(+) calibration. This functionality facilitates a comparison between the different variables of a dataset while considering the effect of the pressure level and the temporal

window. Using this approach, only one variable is assessed at a time, but multiple levels of analogy are possible. Figure 1 results from such an analysis of the NR-1 reanalysis.

### 3.6.5 Global optimization

The sequential calibration has strong limitations: (i) it cannot automatically choose the pressure levels and temporal windows (hour of the day) for a given meteorological variable, (ii) it cannot handle dependencies between parameters, and (iii) it cannot easily handle new degrees of freedom. For this reason, genetic algorithms (GAs) were implemented in AtmoSwing Optimizer to perform a global optimization of AMs. This allows for optimization of all parameters jointly in a fully automatic and objective way. The method is described in Horton et al. (2017a) and an application is provided in Horton et al. (2018).

### 3.6.6 Monte–Carlo simulations

A Monte–Carlo analysis is also implemented in AtmoSwing. The procedure performs thousands of assessments of random parameters within given ranges. This method is not efficient for finding the best parameters set, but facilitates a better understanding of the sensitivity of the parameters. Its relevance is however limited for AMs with multiple levels of analogy and variables. Indeed, for methods with a high number of parameters with wide authorized value ranges, the probability is too low to obtain an acceptable configuration, and thus the resulting response surface might not be representative of the actual distribution of optimal values (See examples in Sect. 4).

## 4 Parameters space of AMs

An analysis of the parameters resulting from Monte–Carlo simulations, the sequential calibration, and GAs was performed for the Binn station in Switzerland (Fig. 1) with ERA-INT (Table 1). High precipitation in the Binn region were responsible for large damages downstream on several occasions, making it a station of particular interest. The results for this station cannot be generalized to all stations, but similar conclusions can be drawn for other locations. Moreover, the parameters space at a single station is expected to be less regular than averaged regional precipitation. For all analyses, the archive period is 1981–2010 and the results are shown for the evaluation period (EP) 2001–2010. For methods requiring a calibration, the calibration period (CP) is 1981–2000. These periods were chosen relatively short to allow for 50,000 Monte–Carlo simulations.

The analysis was first performed for the 2Z method (Table 2). The Monte–Carlo simulations (Fig. 8) show that the spatial window for Z needs to cover a certain region, but can be larger than a critical size. The extent of the spatial window can thus be substantially different without significantly affecting the skill score. This was also observed by Bontron (2004), who noted that "*performance slowly decreases if we consider a window that is slightly too large, while using windows that are too small results in strong performance loss*". The dilution of the relevant synoptic information therefore does not necessarily have a significant negative impact on the skill, while ignoring some of this information leads to stronger losses of skill. The issue of equifinality related to the spatial windows is discussed in Radanovics et al. (2013). The station is usually contained within the optimal spatial domain, provided the predictors are selected for the same day as the predictand. The optimal number of analogs

is relatively well defined, although the selection of more analog candidates is possible without a strong penalty in terms of skill.

The results of the sequential calibration are also illustrated in Fig. 8 (with squares). The calibration was first performed for the CP and applied to EP (blue squares), but was also achieved directly on the EP (red squares). Here, the parameters established on the CP rather than the EP provided slightly better results when assessed on the EP. This is due to the limitations of the sequential calibration that can easily be trapped in a local optima. Indeed, the resulting spatial windows are small in this case, and the algorithm stops as soon as an increase of the domain does not improve the score. This might not be an issue with a low-resolution reanalysis such as NR-1 (2.5°; Table 1), but this might become more of an issue with higher resolutions, such as ERA-INT used here (0.75°), because local minima are more frequent. In this case, the classic+ approach (Sect 3.6.3) might be relevant, but is more time consuming. The Monte-Carlo analysis yielded some better parameter sets than the sequential calibration, due to the constraint on the latter to have the same spatial window for both pressure levels.

Fourteen optimizations by GAs were performed for the same setup (seven optimizations using the classical CP/EP setting (blue triangles) and seven optimizations using the EP as the calibration period directly (red triangles)). The optimization with GAs was given the same degrees of freedom as the Monte–Carlo simulations, so no weighting of the pressure levels was considered (as in Horton et al., 2018). Thus, the parameters optimized for the EP (red) could have been found randomly using the Monte–Carlo simulations. However, this did not occur due to the low probability of obtaining this combination. GAs also result in more skilful parameters than the sequential calibration. When optimized for the EP (red triangles), the parameters yielded results that outperform the optimization for the CP (blue triangles) when assessed on the EP, as it can be expected. Most optimizations converge to a narrow range of values, supposedly, the global optimum for the respective period. The main difference compared to the sequential calibration is that the spatial windows are substantially larger, mainly for Z500, and they differ between pressure levels.

Monte–Carlo simulations were also performed for 2Z-2MI (Table 2) for the same periods and for the same station. Figure 9 shows that the Monte–Carlo simulations could not properly use the moisture variables of the second level of analogy. The boxplots for the second level of analogy show an indifference of the location and the size of the spatial windows, which is demonstrated to be wrong by the sequential calibration and the GAs. Moreover, the achieved CRPS here based on random parameters is not better than the method without the moisture variables (Fig. 8). Additionally, the number of analogs corresponding to the best CRPS values are similar between the two levels of analogy, which means that the second level is simply discarded. There are too many parameters with acceptable ranges that are too narrow to obtain meaningful parameters randomly. Monte–Carlo simulations with uniform probability laws is not suited for even moderately complex AMs.

The sequential calibration results in small spatial windows, especially for moisture variables. The differences with the 2Z methods for the first level of analogy are due to the different initial number of analogs, which has an influence on the choice of the spatial windows. The parameters calibrated for the EP perform better than the ones established on the CP and assessed on the EP, which can be expected.

The results of the GAs show more variability than previously, which is likely due to a higher difficulty related to the larger number of parameters that have to be optimized, and to the presence of potentially more correlated information. The choice of

the spatial windows for the moisture index at 12 h UTC is similar between the different optimization techniques and is a small line of zonally extended points. The chosen spatial windows by GAs for the moisture index at 24 h UTC is surprisingly large. This is likely due to the search of the GAs for additional information at a more distant location due to highly correlated data between 12 h UTC and 24 h UTC at the same 850 hPa level. The lack of convergence for this second spatial window means that the use of this variable is likely not optimal, and it would probably add more information considered at another pressure level, which was shown in Horton et al. (2018). The analysis of the convergence of multiple GA optimizations can thus be useful in interpreting the results and in identifying potentially suboptimal structures or variables.

The former results present a relatively noisy signal for the different optimization methods or the Monte–Carlo simulations. This may be due (1) to the fact that we consider a station's time series instead of regional ones (related to variability from small-scale patterns in the precipitation fields), and (2) because we consider a short period for calibration. Despite the high number of simulations, Monte–Carlo simulations with a uniform probability law are not appropriate for even moderately complex AMs. It is likely that using a Gaussian probability law centered on the station (for the spatial windows) would be more appropriate.

In terms of processing resources, all experiments were done under similar conditions, i.e. using 16 cpus on a Linux cluster. For 2Z, the sequential calibration took 7 min (time is expressed as wall clock time), Monte Carlo took 12.9 h (50,000 evaluations), and GAs took 11.6 h on average (41,000 evaluations on average). For 2Z-2MI, the sequential calibration took 12.5 min, Monte Carlo took 16.8 h, and GAs took 20.4 h on average (61,000 evaluations on average). The computation time should be taken into account in the choice of a calibration strategy.

## 5 Feedback from operational forecast

AtmoSwing Forecaster has been issuing operational forecasts since 2012 for the upper Rhône catchment in Switzerland (Fig. 1) in the context of a flood management project (García Hernández et al., 2009). First, the 2Z and 2Z-2MI methods were implemented using NR-1 as the archive and GFS outputs to describe the target situation (Horton, 2012). Two more recent methods that were optimized by genetic algorithms (Horton et al., 2018) were also implemented since 2016. These methods were found to provide better results both in the perfect prognosis context and in the operational forecast. The results of the forecasts are provided for the Binn station (as in Sect. 4) for the 4Zo method with a lead time of three days (forecast issued three days before the target day; Fig. 10) and the 4Zo-2MIo method with a lead time of one day (Fig. 11). For both methods and both lead times, the forecast obtained by analogs is satisfactory with observations falling within the distribution provided by the analogs. Moreover, when the analog distribution reached high values, it often matched with the observed high precipitation values. As discussed in Sect. 3.4, high precipitation events are better described by the $90^{th}$ percentile than the center of the distribution. The distributions provided by the analogs are quite large but nevertheless provide useful information.

Figure 12 shows how the most significant event (Oct./Nov. 2018) is predicted by the four implemented methods at different lead times. The two older methods (2Z and 2Z-2MI) do not forecast the main peak as well as the optimized ones (4Zo and 4Zo-2MIo). The forecast with a lead time of seven days show that high precipitation amounts can be expected, but the timing is not well defined as the four daily forecasts show high variability of the timing of the occurrence of the peaks (illustrated

by the four forecasted $90^{th}$ percentiles in Fig. 12). The timing and amplitude of the event are relatively well captured by the 4Zo method with a lead time of four days. For the same lead time, adding moisture data (4Zo-2MIo) is not informative as the distributions are wider, and thus the occurrence of the peaks is more uncertain. Globally, moisture data was not very informative for this event. This might be related to the use of NR-1 as the archive, which has a very coarse resolution and was shown to not

perform as well as other reanalyses (Horton and Brönnimann, 2018). It is likely that another dataset would be more accurate, and would be recommended for operational use.

## 6   Limitations of AMs

Although AMs were found to be relevant for several applications, they have some limitations which must be considered. The first is their lower performance for summer compared to winter when using standard predictors (Bliefernicht, 2010). The

relationship between synoptic predictors and local rainfall is lower in the summer, due to convective precipitations that present higher spatial variability and depend on other parameters. The variables that describe the synoptic circulation are indeed not able to predict the location of thunderstorm cells. This was also observed by Ben Daoud (2010), who set up a specific model for the summer months (June 15 to September 15).

Another limitation is the need for a long archive of the predictand variable, for example, precipitation. An alternative for

regions without long archives of station measurements can be using satellite-derived precipitation. Long predictor archives are also required, which is easily satisfied with reanalyses. These may not be perfect in terms of homogeneity, but several can be considered to be of sufficient quality (Horton and Brönnimann, 2018). Moreover, reanalysis data are available all around the world, which represent a great potential for AMs.

Extreme events may be under-represented in the considered sample of analog situations. Indeed, in a limited weather archive,

events with high return periods are not frequent, which can introduce a bias in the prediction. There are however techniques to correct for this bias (see Marty, 2010). In order to produce new extremes, postprocessing of the distribution of analogs might be necessary, for example, by using a scaling based on a predictor variable.

It is also legitimate to raise the question of the relevance of an approach based on archives of past situations in the context of climate change. Changes in circulation frequencies and the persistence of certain weather types (Hewitson and Crane, 1996)

can be accounted for by AMs that contain predictors that characterize atmospheric circulation. Thus, if the archive of weather events is long enough, it is reasonable to assume that a large part of future events is already represented, even those whose frequency will change under different climatic conditions (Wetterhall, 2005). Changes in moisture and temperature variables must be accounted for to correctly capture the climate change signals. Dayon et al. (2015) has demonstrated the transferability of certain AMs to future climate conditions.

## 7 Conclusions

AMs are cost-effective techniques for downscaling local meteorological variables from large-scale predictors. They are used in the context of operational forecasting for flood management or hydropower production, or in a climatic context for climate change impact modelling or reconstruction of past meteorological conditions.

AtmoSwing is a suite of tools that facilitate processing of multiple AM structures in a flexible and efficient way. It consists of four software: the Forecaster for operational forecasting, the Viewer for displaying the Forecaster outputs, the Downscaler for applying AMs in a climatic context, and the Optimizer to establish the relationship between predictors and predictands. AtmoSwing is written in C++, is open source, and has been extensively tested.

Processing operational forecasts with AtmoSwing requires very low computing infrastructure (implementation is possible on a Raspberry Pi 3) yet it can yield useful information, such as early warning for high precipitation events in the case of an application to flood forecasting. Valuable results were obtained in a three-year-long operational forecast in the Swiss Alps. With the global availability of reanalyses, it can be applied to any region with a relatively long predictand time series. The predictors and the structure of the method can be adapted to the local meteorological processes and controlled through xml files. The connection with open access NWP models such as GFS is integrated into AtmoSwing and requires no prior processing. The Forecaster can be installed on a computer or a headless server and run automatically to issue a forecast as soon as new NWP outputs are available. The Viewer offers a user-friendly display of the forecasts, with different levels of synthesis and details. It first provides an overview of potentially critical situations (possibility of high precipitation at a station for a certain lead time) but also allows plotting of the details of the distributions provided by the selected analog dates.

The Downscaler allows the AMs to be used in a climatic context, either for climate reconstruction or for climate change impact studies. When used for future climate analysis, the user must pay close attention to the selected predictors, so that they are able to represent the climate change signal. This is a relatively new field of application of AMs, which was proven to be of interest.

The Optimizer implements different optimization techniques, such as the sequential approach, a Monte–Carlo simulation, and a global optimization technique. Establishing the statistical relationship between predictors and predictand is quite intensive in terms of processing, as it requires numerous assessment over decades. To this end, the Optimizer has been highly optimized in terms of computing efficiency and is parallelized over multiple threads. It scales well on a Linux cluster. This procedure is only required to establish the statistical relationship, which can then be used in forecasting or downscaling at a low computing cost.

One possible key improvement to AtmoSwing Forecaster is a multi-models approach that relies on outputs from multiple global NWP models to better take into account the uncertainty of the NWP forecasts. Similarly, Thevenot (2004) demonstrated the benefit of using ensembles from global NWP as input for the method. The implementation consists of combining the selected analog days associated with each of the members. The forecast on the ensemble was found to be more accurate than the deterministic control for a lead time of four days and more (Thevenot, 2004).

AtmoSwing aims to facilitate implementation of AMs with different types of structure and various predictors while being computationally efficient with low computing requirements. It can be applied to different contexts, such as operational forecasting or climate impact studies. It is open source and will hopefully save future users some development time.

*Code availability.* AtmoSwing is open source and the code can be found in a publicly available GitHub repository (https://github.com/atmoswing/atmoswing). version 2.1.0 is available at https://doi.org/10.5281/zenodo.3208134. The user manual can be found at https://atmoswing.readthedocs.io. AtmoSwing R tools is available at https://doi.org/10.5281/zenodo.1305098. AtmoSwing Python tools is available at https://doi.org/10.5281/zenodo.1495057. The main website for AtmoSwing is http://www.atmoswing.org.

*Author contributions.* P. Horton is the sole developer of AtmoSwing at the time of writing.

*Competing interests.* The authors declare that they have no conflict of interest.

*Acknowledgements.* Thanks to Charles Obled and Michel Jaboyedoff for their valuable scientific inputs during the development of AtmoSwing. Thanks also to Lucien Schreiber and Richard Metzger for their programming advice and to Renaud Marty for his active involvement in testing AtmoSwing against his code base. Thanks also to the two anonymous reviewers, which contributed to improving this paper.

Calculations were performed on UBELIX (http://www.id.unibe.ch/hpc), the HPC cluster at the University of Bern. Precipitation time series were provided by MeteoSwiss. The NCEP/NCAR reanalysis was provided by the NOAA/OAR/ESRL PSD, Boulder, Colorado, USA, at http://www.esrl.noaa.gov/psd/. ERA-interim was obtained from the ECMWF Data Server at http://apps.ecmwf.int/datasets/.

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

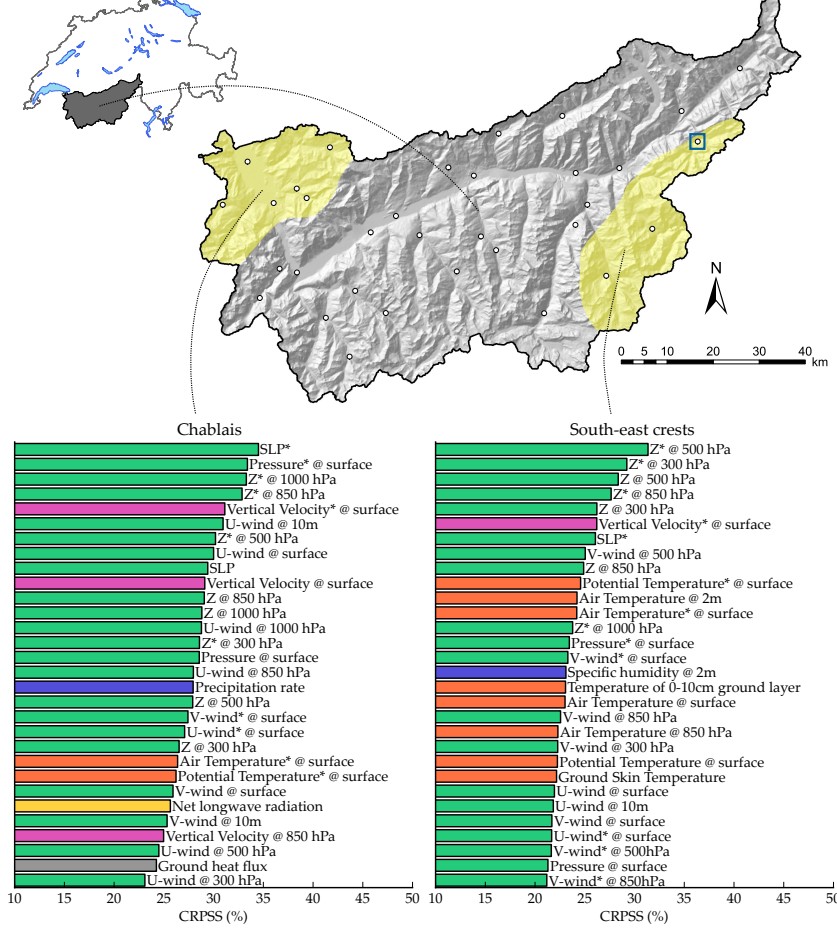

**Figure 1.** Performance score (CRPSS; Eq. 3; the reference being the climatological precipitation distribution) of the 30 best variables from the NR-1 dataset, when considered separately (no combination), for the Chablais region and the southeast ridges in the upper Rhône catchment in Switzerland. The analogy criterion is S1 when an asterisk is present next to the variable name, and RMSE otherwise. Colour illustrates the variable type: green = atmospheric circulation, blue = moisture, orange = temperature, yellow = radiation, purple = vertical velocity, and gray = other. SLP stands for sea level pressure and Z for geopotential height. The blue square indicates the Binn station.

P _ - (A) _ _ _ - (A) _ _ _ ...

Preselection
Type of preselection
Analogy level
Number of predictors
Predictor variable
Optional flag
Analogy level
Number of predictors
Predictor variable
Optional flag

**Figure 2.** Proposed nomenclature to describe the AM structure.

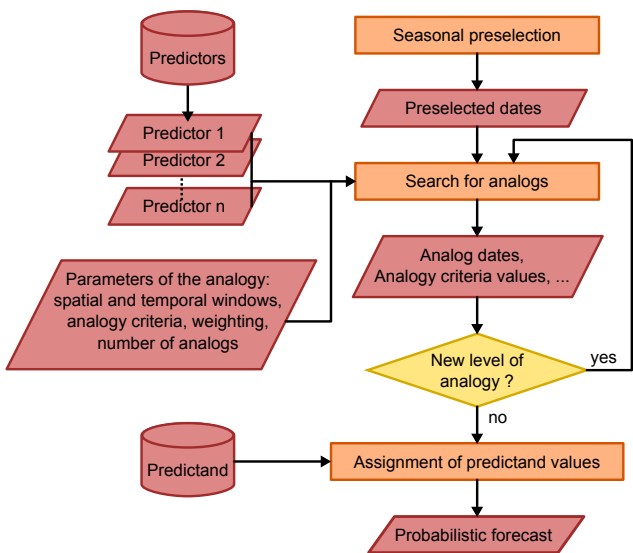

**Figure 3.** Simplified flowchart of the AM implementation in AtmoSwing.

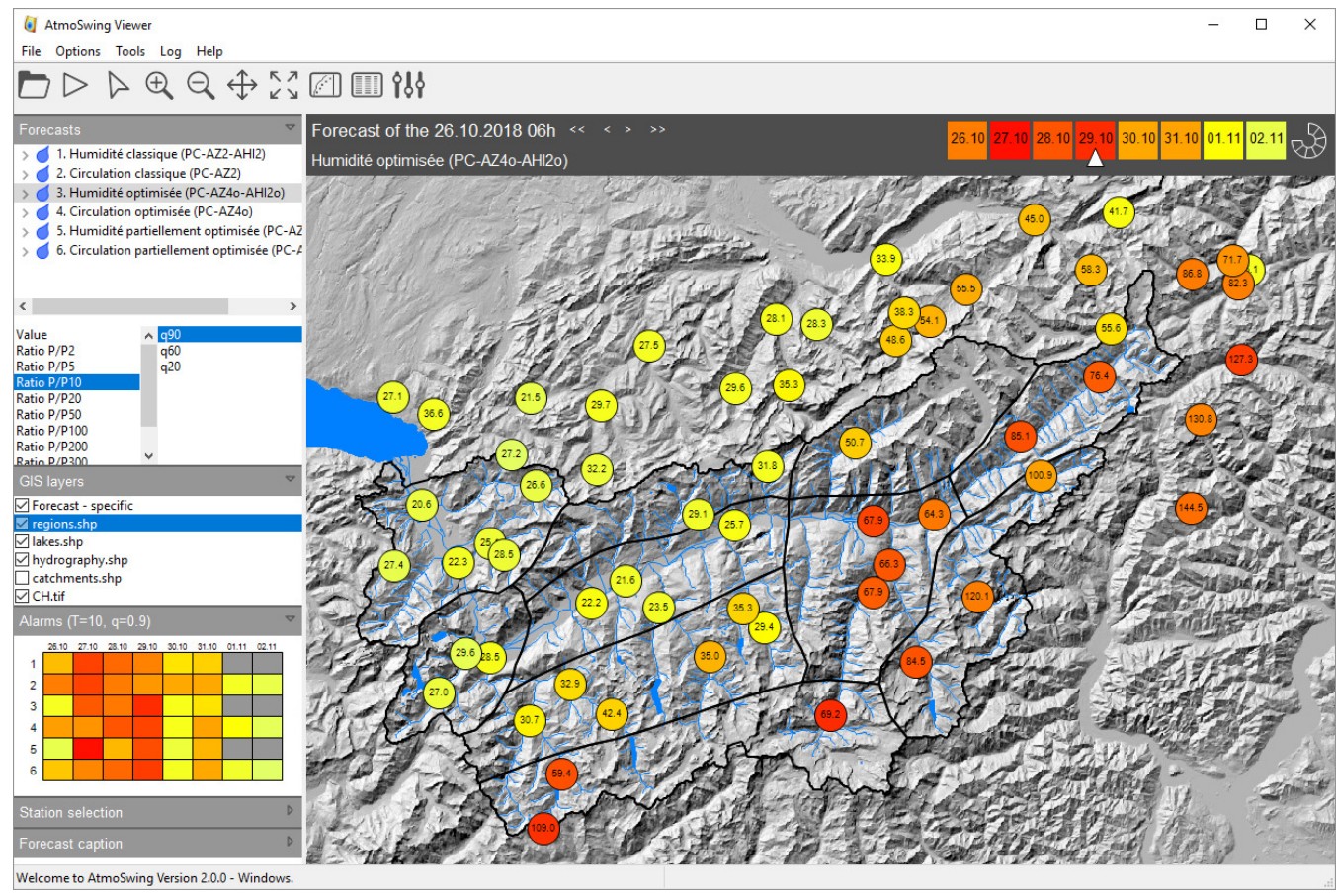

**Figure 4.** Graphical user interface of the Viewer module (Elevation data from The Shuttle Radar Topography Mission - SRTM, and hydrological network from SwissTopo). The values on the map represent the $90^{th}$ percentile (as selected on the left panel) of the precipitation values from the analog samples at the different stations and for the selected lead time. The colour is proportional to the selected return period (10 years here).

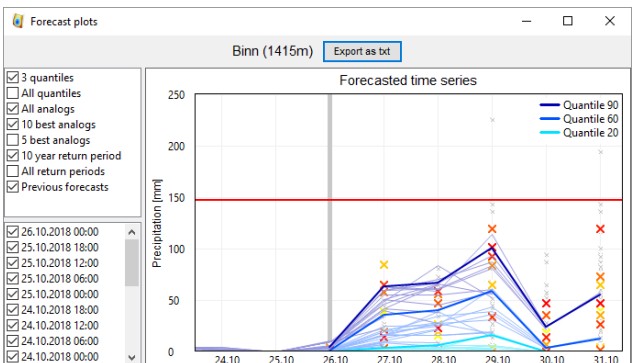

**Figure 5.** Same as Fig. 4 but for multiple lead times.

**Figure 6.** Visualization of the forecasted time series for an event at the Binn station (Fig. 1) in October 2018. The thick blue lines represent the $90^{th}$, $60^{th}$, and $20^{th}$ percentiles for the given lead times. The thin blue lines represent the equivalent time series but from previous forecasts. The small grey crosses represent all analog values and the larger crosses highlight the 10 best analogs (with a colour gradient from red for the best to yellow for the $10^{th}$).

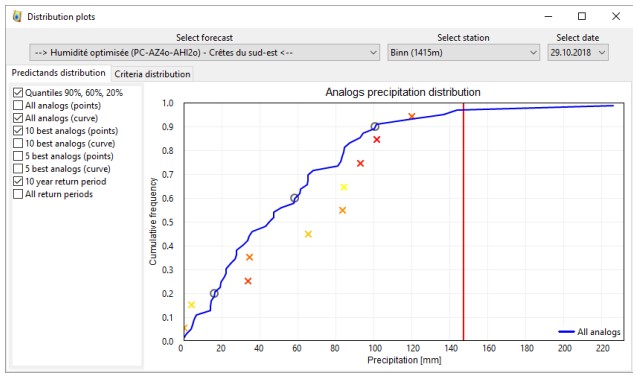

**Figure 7.** Visualization of the forecasted precipitation distribution for a given lead time for an event at the Binn station (Fig. 1) in October 2018. The blue line represent the full distribution provided by all analogs, the circles are the $90^{th}$, $60^{th}$, and $20^{th}$ percentiles, and the crosses correspond to the distribution provided by the 10 best analogs (with a colour gradient from red for the best to yellow for the $10^{th}$). The vertical red line is here the precipitation value for a 10 year return period.

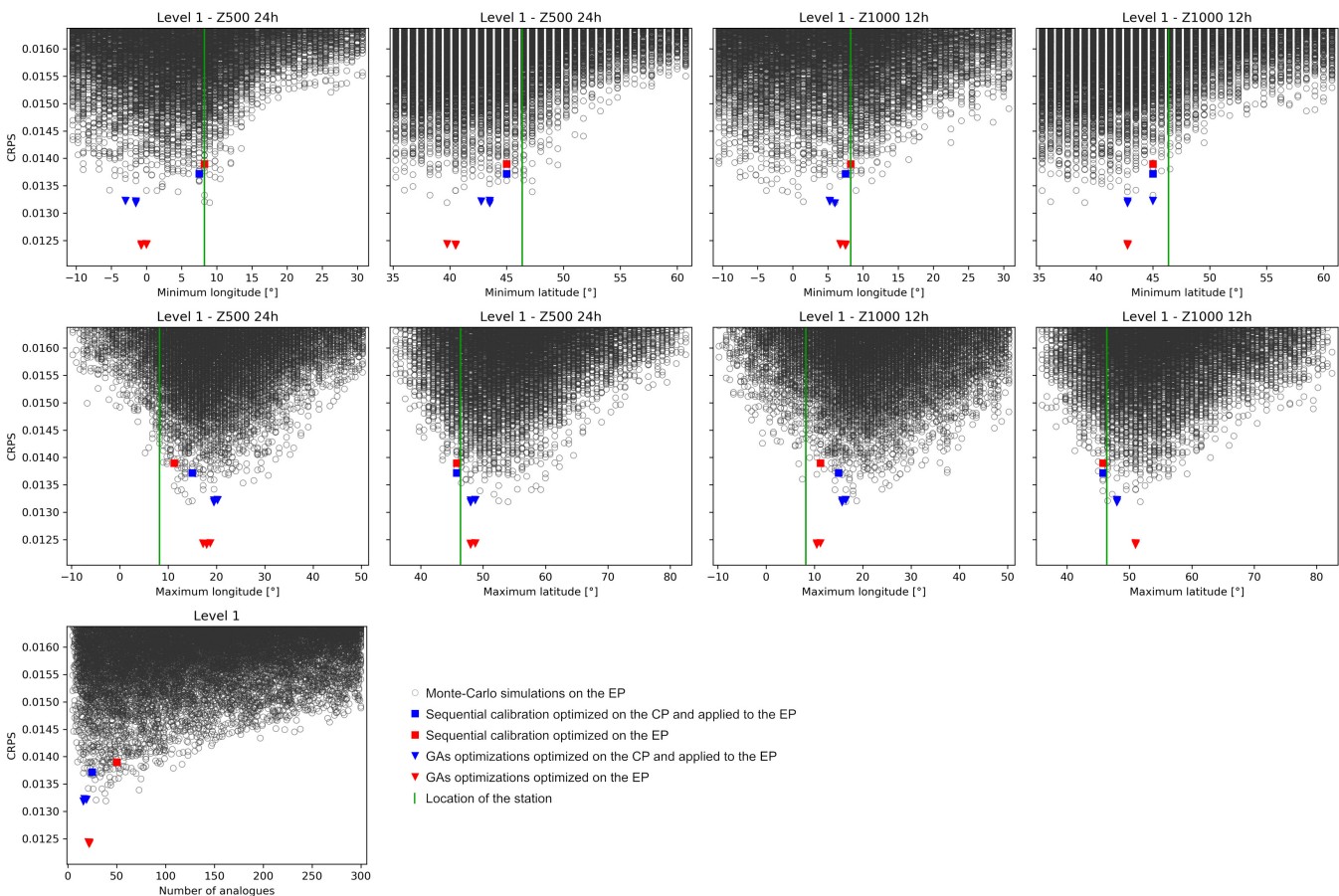

**Figure 8.** Example of parameter values for 2Z (Table 2) for the precipitation at the Binn station (Fig. 1) on the EP. The parameters are the extent (min/max longitude/latitude) of the spatial windows for the geopotential height at 500 and 1000 hPa, and the number of analogs. The green vertical bar in the plots represents the location of the station. The circles represent random parameters from the Monte–Carlo analysis. The plots are truncated at the $25^{th}$ best percentiles for 50,000 realizations. Squares are the results of the sequential calibration and triangles result from genetic algorithms. Markers in blue represent parameters optimized for the CP and applied to EP. Markers in red represent parameters optimized for the EP.

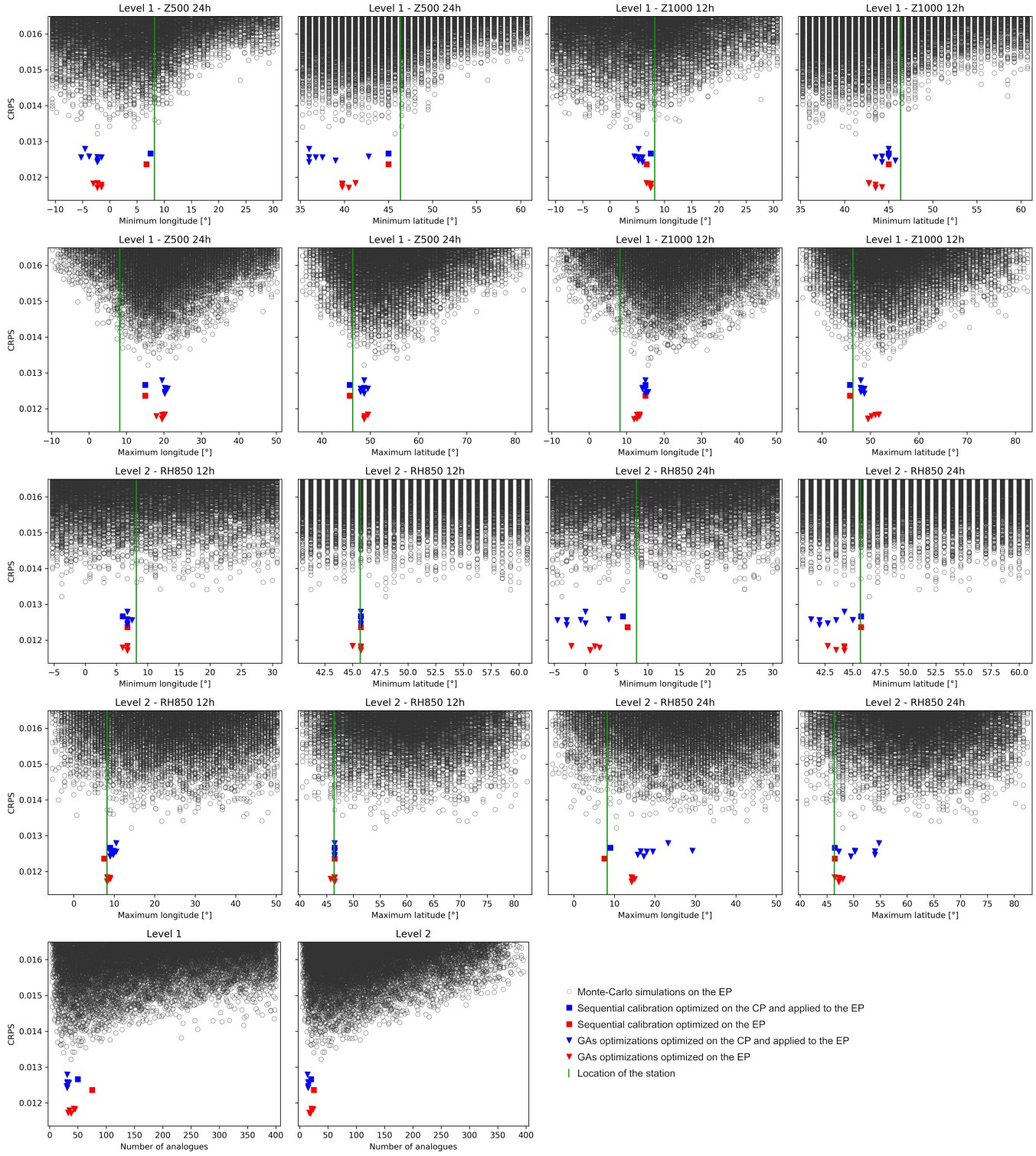

**Figure 9.** Same as Fig. 8 but for 2Z-2MI (Table 2). Results are shown for both levels of analogy (geopotential height and moisture index).

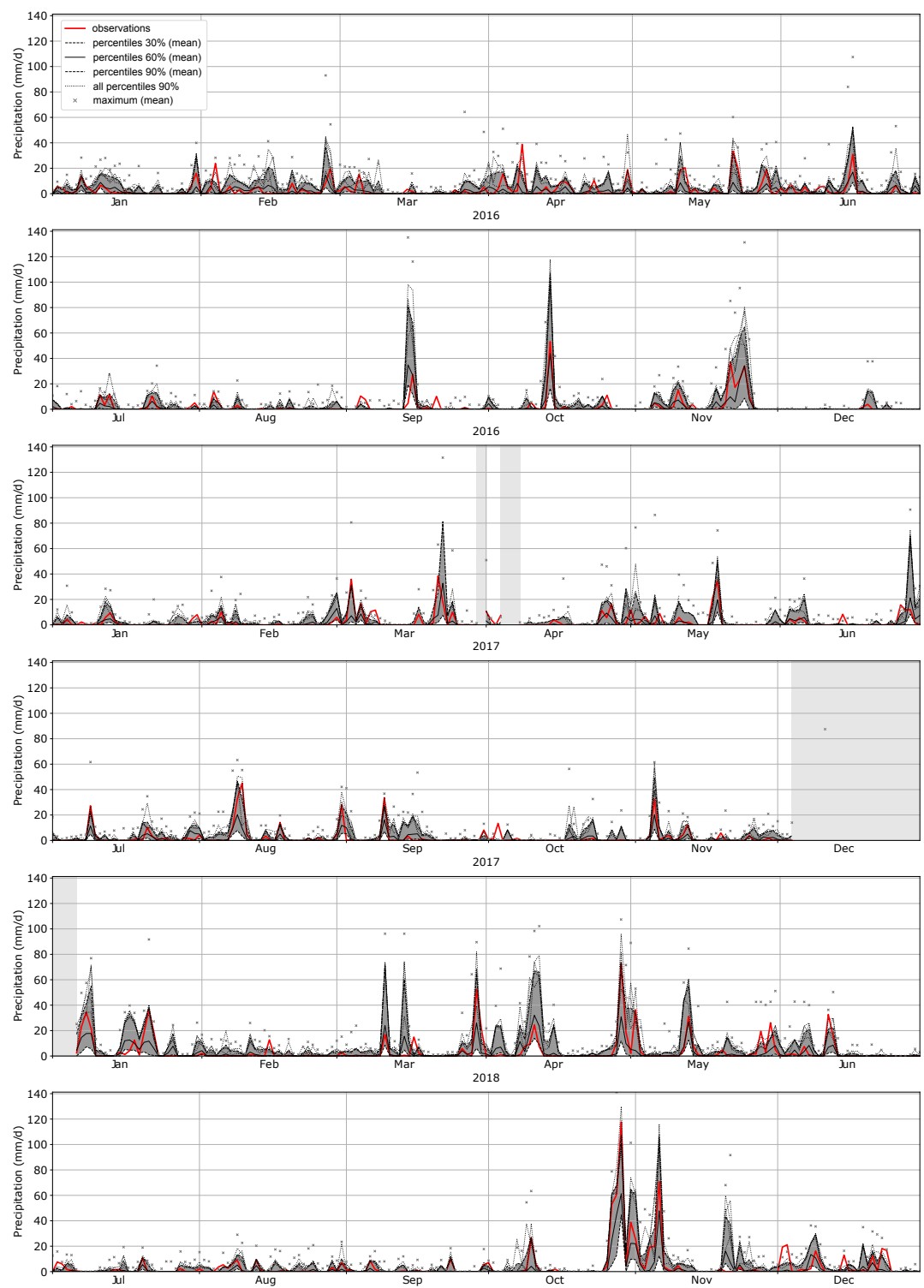

**Figure 10.** Forecasts for the Binn station (Fig. 1) over the period 2016–2018 obtained using the 4Zo method (Table 2) with a lead time of three days. The distributions provided by the analog values are summarized by the $90^{th}$, $60^{th}$, and $30^{th}$ percentiles, as well as the maximum (crosses), all of them averaged over the four daily forecasts. Additionally, the four $90^{th}$ percentiles were also plotted to show the consistency / variability between the four daily forecasts. The shaded areas correspond to forecasts downtime.

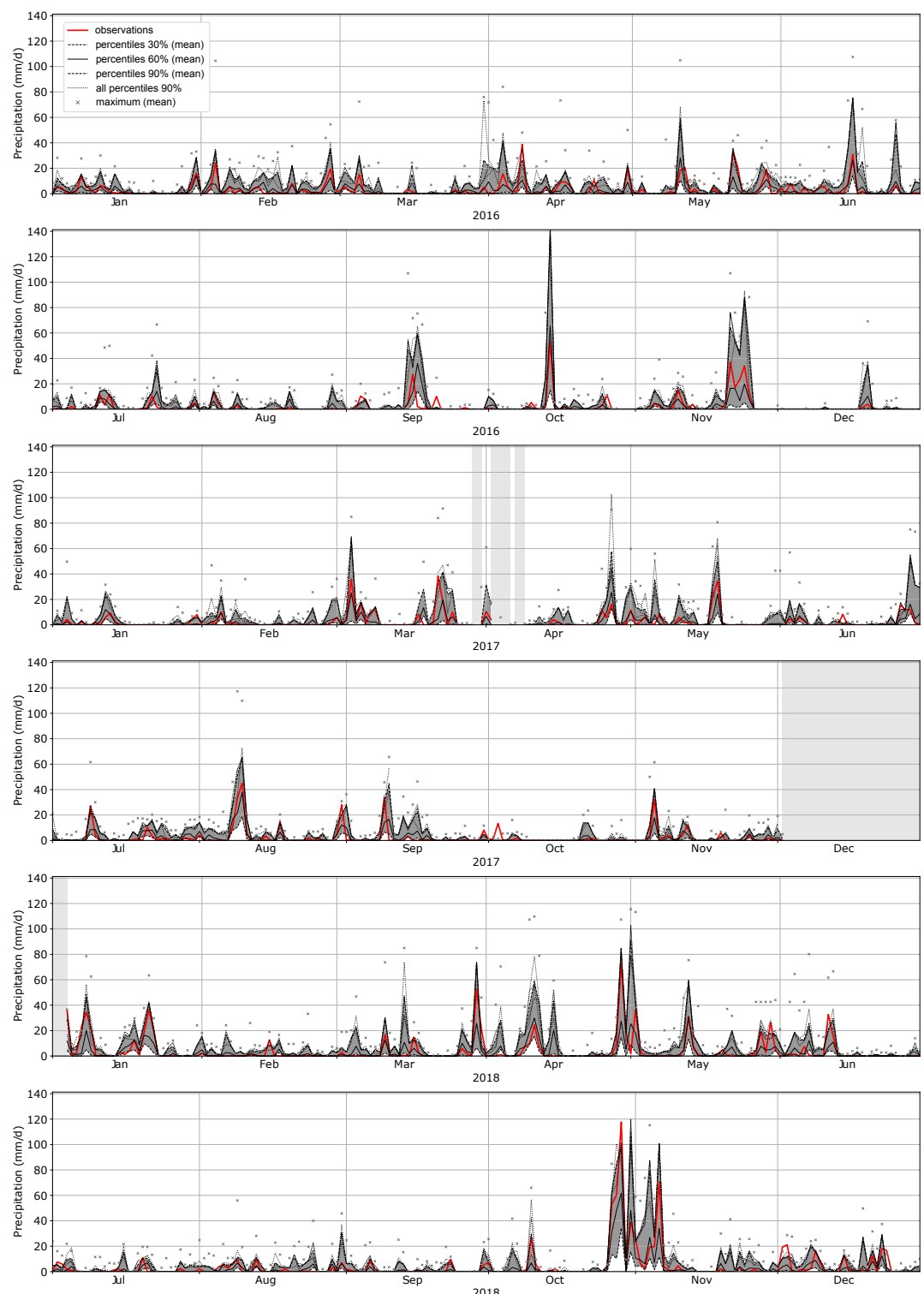

**Figure 11.** Same as Fig. 10 but for the 4Zo-2MIo method (Table 2) with a lead time of one day.

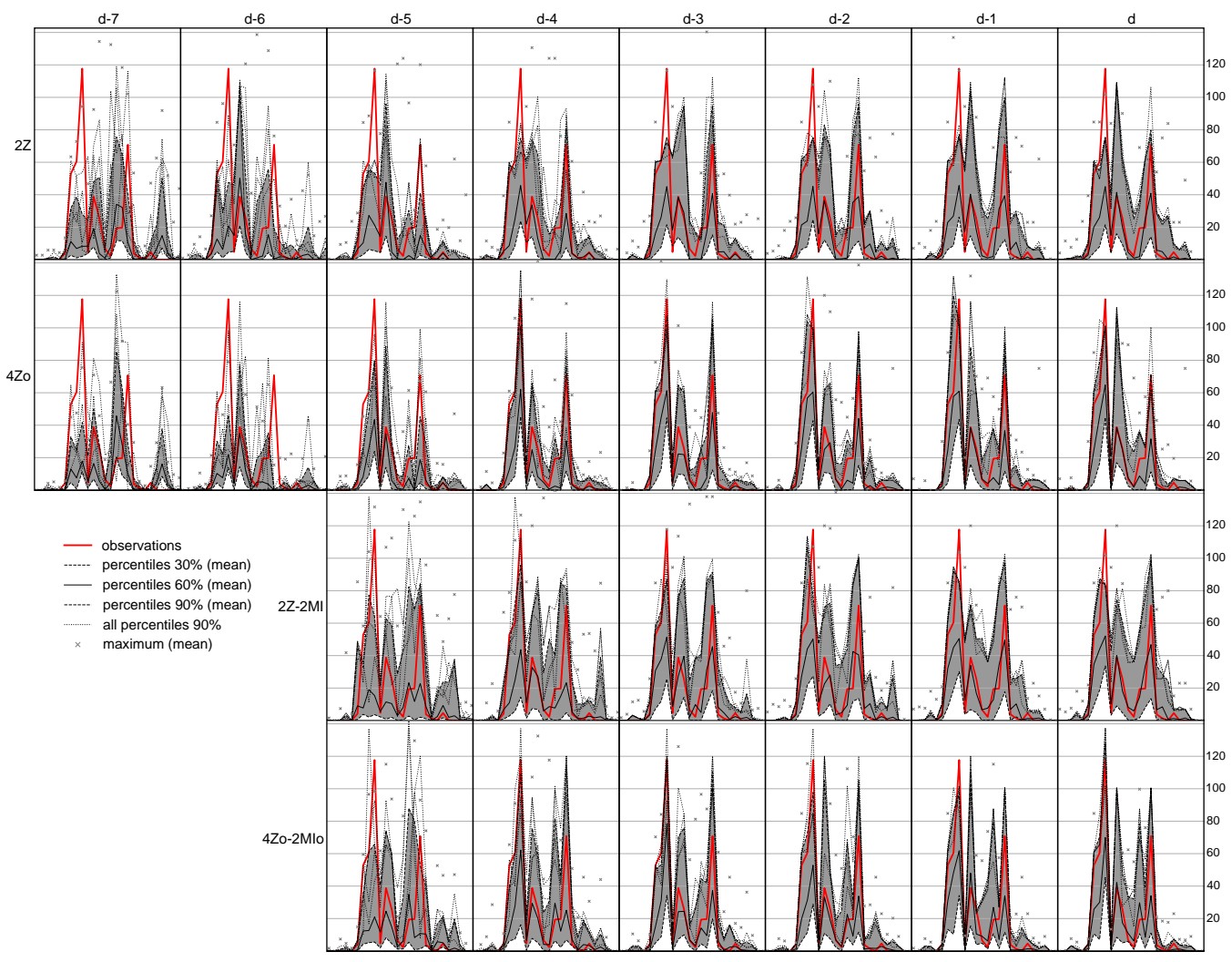

**Figure 12.** Forecasts for the Oct./Nov. 2018 event at the Binn station (Fig. 1) for 2Z, 4Zo, 2Z-2MI and 4Zo-2MIo for lead times from seven to zero days prior to the target day.

**Table 1.** Reanalysis datasets that can be read by AtmoSwing.

| Name | Institution | Period of record | Output resolution | Model resolution | Model generation | Type of input |
|---|---|---|---|---|---|---|
| NR-1 | NCEP, NCAR | 1948 – present | 2.5°x 2.5° | T62 ($\sim$1.88°), L28 | 1995 | full |
| NR-2 | NCEP, DOE | 1979 – present | 2.5°x 2.5° | T62 ($\sim$1.88°), L28 | 2001 | full |
| ERA-INT | ECMWF | 1979 – present | 0.75°x 0.75° | TL255 ($\sim$0.70°), L60 | 2006 | full |
| 20CR-2c | NOAA-CIRES | 1851 – 2014 | 2°x 2° | T62 ($\sim$1.88°), L28 | 2008 | surface |
| CFSR | NCEP | 1979 – present | 0.5°x 0.5° | T382 ($\sim$0.31°), L64 | 2009 | full |
| JRA-55 | JMA | 1958 – present | 1.25°x 1.25° | TL319 ($\sim$0.36°), L60 | 2009 | full |
| JRA-55C | JMA | 1958 – 2015 | 1.25°x 1.25° | TL319 ($\sim$0.36°), L60 | 2009 | conventional |
| ERA-20C | ECMWF | 1900 – 2010 | 1°x 1° | TL159 ($\sim$1.13°), L91 | 2012 | surface |
| MERRA-2 | NASA GMAO | 1980 – present | 0.625°x 0.5° | 0.625°x 0.5°, L72 | 2014 | full |
| CERA-20C | ECMWF | 1901 – 2010 | 1°x 1° | T159 ($\sim$1.13°), L91 | 2016 | surface |
| ERA5 | ECMWF | 1979 – present | 0.25°x 0.25° | TL639 ($\sim$0.28°), L137 | 2016 | full |

**Table 2.** Some existing analog methods, listed by increasing complexity. P0 is the preselection (PC: on calendar basis, that is ±60 days around the target date), L1, L2 and L3 are the subsequent levels of analogy. N1, N2 and N3 are the number of analogs to select at each level of analogy. The meteorological variables are: Z – geopotential height, T – air temperature, W – vertical velocity, MI – moisture index, which is the product of the relative humidity at the given pressure level and the total water column, MF – moisture flux, which is the product of MI with the wind intensity. The analogy criterion is S1 for Z and RMSE for the other variables.

| Type | P0 | L1 | N1 | L2 | N2 | L3 | N3 | Reference |
|------|-----|------|------|------|------|------|------|-----------|
| **2Z** | PC | Z1000@12h | 50 | | | | | Bontron 2004 |
| | | Z500@24h | | | | | | |
| **4Z** | PC | Z1000@06h | ~27 | | | | | Horton et al. 2018 |
| | | Z1000@30h | | | | | | |
| | | Z700@24h | | | | | | |
| | | Z500@12h | | | | | | |
| **2Z-2MI** | PC | Z1000@12h | 70 | MI850@12h | 30 | | | Bontron 2004 |
| | | Z500@24h | | MI850@24h | | | | |
| **2Z-2MI** | PC | Z1000@06h | 75 | MI925@06h | 30 | | | Marty 2010 |
| | | Z500@18h | | MI925@18h | | | | |
| **2Z-2MF** | PC | Z1000@06h | 60 | MF700@06h[†] | 25 | | | Marty 2010 |
| | | Z500@18h | | MF700@18h | | | | |
| **4Z-2MI** | PC | Z1000@30h | ~63 | MI700@24h | ~24 | | | Horton et al. 2018 |
| | | Z850@12h | | MI600@12h | | | | |
| | | Z700@24h | | | | | | |
| | | Z400@12h | | | | | | |
| **PT-2Z-4MI** | T925@36h | Z1000@12h | 70 | MI925@12h | 25 | | | Ben Daoud et al. 2016 |
| | T600@12h | Z500@24h | | MI925@24h | | | | |
| | | | | MI700@12h | | | | |
| | | | | MI700@24h | | | | |
| **PT-2Z-10MI** | T925@36h | Z1000@12h | 70 | MI925@06-30h | 25 | | | Ben Daoud 2010 |
| | T600@12h | Z500@24h | | MI700@06-30h | | | | |
| **PT-2Z-4W-4MI** | T925@36h | Z1000@12h | 170 | W850@06h | 70 | MI925@12h | 25 | Ben Daoud et al. 2016 |
| | T600@12h | Z500@24h | | W850@12h | | MI925@24h | | |
| | | | | W850@18h | | MI700@12h | | |
| | | | | W850@24h | | MI700@24h | | |

† or MF925@06h+18h as an alternative