# Peer review of "AtmoSwing: Analog Technique Model for Statistical Weather forecastING and downscalING (v2.1.0)"

_Geoscientific Model Development, 2019_

## Short Comment (SC1) · 24 Apr 2019

Dear authors,

In my role as Executive editor of GMD, I would like to bring to your attention our Editorial version 1.1:

http://www.geosci-model-dev.net/8/3487/2015/gmd-8-3487-2015.html

This highlights some requirements of papers published in GMD, which is also available on the GMD website in the 'Manuscript Types' section:

http://www.geoscientific-model-development.net/submission/manuscript_types.html

In particular, please note that for your paper, the following requirements have not been

met in the Discussions paper:

- "The main paper must give the model name and version number (or other unique identifier) in the title."

- "All papers must include a section, at the end of the paper, entitled 'Code availability'. Here, either instructions for obtaining the code, or the reasons why the code is not available should be clearly stated. It is preferred for the code to be uploaded as a supplement or to be made available at a data repository with an associated DOI (digital object identifier) for the exact model version described in the paper. Alternatively, for established models, there may be an existing means of accessing the code through a particular system. In this case, there must exist a means of permanently accessing the precise model version described in the paper. In some cases, authors may prefer to put models on their own website, or to act as a point of contact for obtaining the code. Given the impermanence of websites and email addresses, this is not encouraged, and authors should consider improving the availability with a more permanent arrangement. After the paper is accepted the model archive should be updated to include a link to the GMD paper."

Thus, add a version number of AtmoSwing to the title. We very much appreciate, that the code is fully freely available. However, to fully meet the two criteria to archive the **exact** code version **permanently** please provide a DOI for the exact code version published in this article. Note that for projects in GitHub (such as AtmoSwing) a DOI can easily be created using Zenodo, see https://guides.github.com/activities/citable-code/ for details.

Yours,

Astrid Kerkweg

---

## Author Comment (AC1) · 26 Apr 2019

Dear executive editor,

Thanks for your comment. With regard to the first point, we omitted the version number on purpose to avoid confusion from the reader. Indeed, the software version number is 2.0, while this is the first article about the model and it is meant to be the reference paper. We were afraid people would be looking for an article for version 1. Furthermore, we saw several other recent articles in GMD without version number in the title. However, if we have to, we will add it to the title.

About the 'Code availability' section, there is one before the Acknowledgements. There is also a version number and a DOI generated by Zenodo, but in the references (Hor-

ton, 2018a): Horton, P.: AtmoSwing v2.0.0, https://doi.org/10.5281/zenodo.1491744, 2018a. We can move this information from the references to the Code availability section for more clarity.

Best regards,

Pascal Horton

---

## Referee Comment (RC1) · Anonymous Referee #1 · 10 May 2019

General comments:

This paper presents AtmoSwing, an open source software that allows implementation of Analog Methods (AMs), from calibration to operational use. It can be used either in the field of forecasting (typically rainfall forecasting) or in that of climate studies. Being very versatile, this software might prove very useful for operational uses as well as for academic studies. Furthermore, this paper has a high educational interest as it presents a comprehensive bibliographic review of the AMs development, defining this way clearly the AMs state-of-the-art.

Specific comments:

P1 L1: I would not use the term "prediction" as, in my point of view, the AMs are not

forecasting methods by themselves, but rather adaptation methods, which link predictand to predictors (as it is well explained by the author himself p.7, l6-25).

P2. L7-8: "(...) one describing the situation (...)" you should specify that these are 'historical situations' that will be compared to the situation at hand.

P3 L23-13 : I do not agree to the terms "partially independent forecast". As express before, the AMs are not forecasting methods. The forecasting capacity is due to the NWP. The AMs are adapatation methods that can enhance the forecasting skill of the NWP.

P4 L20-21 : these results were obtained considering daily rainfall (for shorter time-step, we may assume that we could use shorter archives).

P7 L31 : Indicate that the CRPSS score used in Fig. 1 is explained section 3.6.2. Indicate also what is the reference forecast used to compute the CRPSS

P15 equation (2) : I think the subscript i of H must be removed.

P17. Section 4 : All this section is very interesting. Is it possible to add the computing time requested by the Monte-Carlo simulation, the sequential calibration and the GAs calibration, for each case? The comparison of these computing times with the obtained skills might be quite interesting.

Technical corrections:

none
* * *

---

## Referee Comment (RC2) · Anonymous Referee #2 · 22 May 2019

**1 Summary**

The paper presents the AtmoSwing software. AtmoSwing can be used to optimise and apply analogue models in various contexts and to visualize the results in an interactive way. The software is the fruit of many years of research and development work conducted by the author.

[Figure]

**2   General comments**

I appreciate the sound development work presented in the paper. The paper is scientifically sound, well structured and merits publication. However, there are a few minor points to be clarified.

**3   Specific comments**

1. page 2 line 30-33: The literature cited here is very old. Are analogue methods still used for the cited purposes? If this is the case, please replace with more recent literature, if not, I think this paragraph could be omitted. If you decide to keep it, please marque as historical use.

2. page 3 line 1-5: Please mention the predictands evaluated in the mentioned project.

3. page 5 line 18: "This preselection is now often implemented as a moving selection..." Please add a reference.

4. page 5 line 28 and page 19 line 13-14: I don't understand the hours UTC here. Especially 24h UTC, that would be rather 00h UTC. What is the reasoning behind taking values at a specific time of day? Does the choice depend on the longitude? Or are those hours meant to be forecast lead-times or time ranges? In this case "UTC" doesn't make any sense. Please clarify.

5. page 6 line 10: You state that the moisture index MI "does not represent an actual physical quantity, but expresses the water content and the degree of saturation". To me the water content and the degree of saturation are physical quantities. Please clarify.

6. page 6 line 21: What does "close in distance but too dissimilar in pattern" mean? Doesn't a distance in PCA space measure dissimilarity in the contribution of different patterns?

7. page 6 line 26: Similarly, "an analogy of the atmospheric circulation instead of a Euclidean distance" is not clear to me.

8. page 6 line 27: Isn't the RMSE the same as the Euclidean distance in this case?

9. page 7 line 16-25: The term "temporal extrapolation" is confusing in this paragraph. It makes me think of techniques like kinematic extrapolation which are used for example in a nowcasting context. I understood that in the first part of the paragraph you talk about analogy of temporal trajectories and their limitations. In the second part NWP forecasts are used on the synoptic scale, but they are based on the numerical resolution of dynamic equations and not extrapolation. Please revise.

10. page 7 line 22: You mention precipitation and temperature as examples for predictands that are difficult to simulate for numerical models. I'm not sure that temperature is a very good example here, given the performance of modern weather forecast models. What do you think? Under which circumstances and for which temporal scales an analogue forecast of temperature typically performs better than a numerical model?

11. page 12 line 23: "Different authors" which ones?

12. page 16 line 25: I think it would be useful to specify which kinds of objective functions are minimized and which ones are maximized. For example error functions are minimized and skill scores are maximized.

13. page 18: At some point I got a bit lost between "calibration periods", "optimisation periods", "archive periods" and "validation periods". Please define calibration

period vs. optimisation period. In which cases within sample skill is measured, and in which cases out of sample skill is measured? Is the archive length always the same? In line 29: "The contrary is expected for the later period" please explain why.

14. page 19 line 10: Why is this expected? Please specify.

15. figure 4: This figure is unclear to me. Especially the meaning of the connections with different line types, arrows and points. I didn't look at the code, so the figure might be useful in the software documentation or user manual, but I don't see the purpose of this figure within the paper.

16. figure 5: What are the numbers in the circles? Please add the information in the caption.

17. figure 7 and 8: Incomplete caption. What are the crosses?

18. figure 9 caption: What does "optimised directly" mean?

19. figure 11, 12 and 13: The axis annotations, legends and crosses are very small. Please increase their size.

**4  Technical corrections**

1. page 2 lines 9-13: I'd suggest to stick to present tense here.

2. page 2 lines 15-17: There is some wording in this sentence that seems strange to me in the given context: Incidentally, proposed, assumed, most efficient. Please rephrase the sentence.

3. page 3 line 29: There are sections missing in the outline.

4. page 4 lines 30-31: These phrases are a bit vague, please rephrase.

5. page 5 line 9: "that which" -> "the one that"

6. page 5 line 14: "eventual" -> Did you mean "possible", "potential" or "optional" ?

7. page 5 line 15: "compromise to take into account" I'd suggest "trade-off between taking into account..."

8. page 5 line 24: determined -> found?

9. page 5 line 26: "plays a greater significance" -> "plays a greater role"

10. page 7 line 9: Paranthesis around the citations are missing.

11. page 7 line 29: "catchment" is used here while "region" is used before and after.

12. page 7 line 29: "for use in" -> "for their use in"

13. page 17 line 28: The Binn station was responsible for damages? Please rephrase.

14. page 22 line 4: "allowing them to be" -> "such as"

15. figure 12: Typo in the caption: Fig. 12 -> Fig. 11

16. table 1: "model vintage" -> "model generation" (vintage is rather for wine)

---

## Author Response (AR1)

**Reply to reviewer 1**

Thanks to the reviewer for his positive review and his constructive comments.

**Comment 1:**

P1 L1: I would not use the term "prediction" as, in my point of view, the AMs are not forecasting methods by themselves, but rather adaptation methods, which link predictand to predictors (as it is well explained by the author himself p.7, l6-25).

Reply: The first sentence has been changed, as well as other uses of the term "prediction".

**Comment 2:**

P2. L7-8: "(…) one describing the situation (…)" you should specify that these are 'historical situations' that will be compared to the situation at hand.

Reply: Thanks for the suggestion; this was added.

**Comment 3:**

P3 L23-13 : I do not agree to the terms "partially independent forecast". As express before, the AMs are not forecasting methods. The forecasting capacity is due to the NWP. The AMs are adapatation methods that can enhance the forecasting skill of the NWP.

Reply: I do agree with reviewer 1, and this was the meaning of "partially". However, to avoid confusion, this has been changed to "statistical adaptation".

**Comment 4:**

P4 L20-21 : these results were obtained considering daily rainfall (for shorter time-step, we may assume that we could use shorter archives).

Reply: Thanks for the suggestion; it was added to the manuscript.

**Comment 5:**

P7 L31 : Indicate that the CRPSS score used in Fig. 1 is explained section 3.6.2. Indicate also what is the reference forecast used to compute the CRPSS

Reply: This has been added to the caption of Fig. 1.

**Comment 6:**

P15 equation (2) : I think the subscript i of H must be removed.

Reply: Correct, thanks for identifying this.

**Comment 7:**

P17. Section 4 : All this section is very interesting. Is it possible to add the computing time requested by the Monte-Carlo simulation, the sequential calibration and the GAs calibration, for each case? The comparison of these computing times with the obtained skills might be quite interesting.

Reply: The following paragraph was added: "In terms of processing resources, all experiments were done under similar conditions, i.e. using 16 cpus on a Linux cluster. For 2Z, the sequential calibration

took 7 min (time is expressed as wall clock time), Monte Carlo took 12.9 h (50,000 evaluations), and GAs took 11.6 h on average (41,000 evaluations on average). For 2Z-2MI, the sequential calibration took 12.5 min, Monte Carlo took 16.8 h, and GAs took 20.4 h on average (61,000 evaluations on average). The computation time should be taken into account in the choice of a calibration strategy."

**Reply to reviewer 2**

Thanks to the reviewer for his thoughtful and detailed comments. All technical corrections were addressed and will not be discussed here.

**Comment 1:**

page 2 line 30-33: The literature cited here is very old. Are analogue methods still used for the cited purposes? If this is the case, please replace with more recent literature, if not, I think this paragraph could be omitted. If you decide to keep it, please marque as historical use.

Reply: This literature has been updated.

**Comment 2:**

page 3 line 1-5: Please mention the predictands evaluated in the mentioned project.

Reply: This has been added (for daily precipitation).

**Comment 3:**

page 5 line 18: "This preselection is now often implemented as a moving selection..." Please add a reference.

Reply: Some references were added.

**Comment 4:**

page 5 line 28 and page 19 line 13-14: I don't understand the hours UTC here. Especially 24h UTC, that would be rather 00h UTC. What is the reasoning behind taking values at a specific time of day? Does the choice depend on the longitude? Or are those hours meant to be forecast lead-times or time ranges? In this case "UTC" doesn't make any sense. Please clarify.

Reply: The following sentence was added in parenthesis to explain the selection of hours: ''reference time of the predictors as they are usually available at a 6-hrly temporal resolution or higher''. 24h UTC means here at 00h UTC but the next day. The reason being that daily precipitation is usually measured between 6h UTC and 6h UTC the next day, and so the centre of the accumulation period is 18h UTC. A couple of predictors at 12h UTC and 24h UTC is then centred on the accumulation period.

**Comment 5:**

page 6 line 10: You state that the moisture index MI "does not represent an actual physical quantity, but expresses the water content and the degree of saturation". To me the water content and the degree of saturation are physical quantities. Please clarify.

Reply: This sentence has been removed.

**Comment 6:**

page 6 line 21: What does "close in distance but too dissimilar in pattern" mean? Doesn't a distance in PCA space measure dissimilarity in the contribution of different patterns?

Reply: This sentence has been removed as it is out of the scope of the paper anyway.

**Comment 7:**

page 6 line 26: Similarly, "an analogy of the atmospheric circulation instead of a Euclidean distance" is not clear to me.

Reply: The sentence has been changed to: '' S1 allows for a comparison of the gradients and thus an analogy of the atmospheric circulation instead of considering the actual values at the grid points''.

**Comment 8:**

page 6 line 27: Isn't the RMSE the same as the Euclidean distance in this case?

Reply: Yes. The sentence is now: ''For other predictors, classic criteria representing Euclidean distances between grid point values are used: …''.

**Comment 9:**

page 7 line 16-25: The term "temporal extrapolation" is confusing in this paragraph. It makes me think of techniques like kinematic extrapolation which are used for example in a nowcasting context. I understood that in the first part of the paragraph you talk about analogy of temporal trajectories and their limitations. In the second part NWP forecasts are used on the synoptic scale, but they are based on the numerical resolution of dynamic equations and not extrapolation. Please revise.

Reply: The paragraph has been edited and is now: '' In one of the very first uses in operational forecasting, radiosonde observations were used as predictors to predict precipitation for the next two days. However, because of the chaotic nature of the atmosphere, two analog situations quickly diverge over time (Lorenz, 1969). Thus, the AM has strong limitations regarding the analogy of temporal trajectories (Bontron, 2004). Given the superior capability of numerical models for simulating the dynamic evolution of the atmosphere, their outputs are now used as predictors for the coming days. …''

**Comment 10:**

page 7 line 22: You mention precipitation and temperature as examples for predictands that are difficult to simulate for numerical models. I'm not sure that temperature is a very good example here, given the performance of modern weather forecast models. What do you think? Under which circumstances and for which temporal scales an analogue forecast of temperature typically performs better than a numerical model?

Reply: That was not intentional, but the result of consecutive editions. The sentence has been changed to ''… with a local predictand, especially precipitation, which is more difficult to simulate for numerical models.''

**Comment 11:**

page 12 line 23: "Different authors" which ones?

Reply: They are actually listed at the end of the sentence: (Djerboua, 2001; Bontron, 2004; Marty, 2010)

**Comment 12:**

page 16 line 25: I think it would be useful to specify which kinds of objective functions are minimized and which ones are maximized. For example error functions are minimized and skill scores are maximized.

Reply: Thanks for the suggestion. This has been added.

**Comment 13:**

page 18: At some point I got a bit lost between "calibration periods", "optimisation periods", "archive periods" and "validation periods". Please define calibration period vs. optimisation period. In which cases within sample skill is measured, and in which cases out of sample skill is measured? Is the archive length always the same? In line 29: "The contrary is expected for the later period" please explain why.

Reply: The section has been edited and a paragraph was added in the beginning: '' For all analyses, the archive period is 1981-2010 and the results are shown for the evaluation period (EP) 2001-2010. For methods requiring a calibration, the calibration period (CP) is 1981-2000.'' The sentence "The contrary is expected for the later period" has been removed as it is not of primary importance. A visual caption has also been added to the figure.

**Comment 14:**

page 19 line 10: Why is this expected? Please specify.

Reply: The sentence has be rephrased: '' The parameters calibrated for the EP perform better than the ones established on the CP and assessed on the EP, which can be expected.''

**Comment 15:**

figure 4: This figure is unclear to me. Especially the meaning of the connections with different line types, arrows and points. I didn't look at the code, so the figure might be useful in the software documentation or user manual, but I don't see the purpose of this figure within the paper.

Reply: The figure has been removed.

**Comment 16:**

figure 5: What are the numbers in the circles? Please add the information in the caption.

Reply: The caption has been completed: The values on the map represent the 90th percentile (as selected on the left panel) of the precipitation values from the analogs at the different stations and for the selected lead time. The colour is proportional to the selected return period (10 years here).

**Comment 17:**

figure 7 and 8: Incomplete caption. What are the crosses?

Reply: This has been added for figure 7: "The thick blue lines represent the 90th, 60th, and 20th percentiles for the given lead times. The thin blue lines represent the equivalent time series but from previous forecasts. The small grey crosses represent all analog values and the larger crosses highlight the 10 best analogs (with a colour gradient from red for the best to yellow for the 10th)."

For figure 8: "The blue line represent the full distribution provided by all analogs, the circles are the 90th, 60th, and 20th percentiles, and the crosses correspond to the distribution provided by the 10 best analogs (with a colour gradient from red for the best to yellow for the 10th). The vertical red line is here the precipitation value for a 10 year return period."

**Comment 18:**

figure 9 caption: What does "optimised directly" mean?

Reply: "directly" has been removed.

**Comment 19:**

figure 11, 12 and 13: The axis annotations, legends and crosses are very small. Please increase their size.

Reply: This has been fixed.

[revised manuscript text omitted]

**Table 2.** Some existing analog methods, listed by increasing complexity. P0 is the preselection (PC: on calendar basis, that is ±60 days around the target date), L1, L2 and L3 are the subsequent levels of analogy. N1, N2 and N3 are the number of analogs to select at each level of analogy. The meteorological variables are: Z – geopotential height, T – air temperature, W – vertical velocity, MI – moisture index, which is the product of the relative humidity at the given pressure level and the total water column, MF – moisture flux, which is the product of MI with the wind intensity. The analogy criterion is S1 for Z and RMSE for the other variables.

| Type | P0 | L1 | N1 | L2 | N2 | L3 | N3 | Reference |
|---|---|---|---|---|---|---|---|---|
| 2Z | PC | Z1000@12h Z500@24h | 50 | | | | | Bontron 2004 |
| 4Z | PC | Z1000@06h Z1000@30h Z700@24h Z500@12h | ~27 | | | | | Horton et al. 2018 |
| 2Z-2MI | PC | Z1000@12h Z500@24h | 70 | MI850@12h MI850@24h | 30 | | | Bontron 2004 |
| 2Z-2MI | PC | Z1000@06h Z500@18h | 75 | MI925@06h MI925@18h | 30 | | | Marty 2010 |
| 2Z-2MF | PC | Z1000@06h Z500@18h | 60 | MF700@06h[†] MF700@18h | 25 | | | Marty 2010 |
| 4Z-2MI | PC | Z1000@30h Z850@12h Z700@24h Z400@12h | ~63 | MI700@24h MI600@12h | ~24 | | | Horton et al. 2018 |
| PT-2Z-4MI | T925@36h T600@12h | Z1000@12h Z500@24h | 70 | MI925@12h MI925@24h MI700@12h MI700@24h | 25 | | | Ben Daoud et al. 2016 |
| PT-2Z-10MI | T925@36h T600@12h | Z1000@12h Z500@24h | 70 | MI925@06-30h MI700@06-30h | 25 | | | Ben Daoud 2010 |
| PT-2Z-4W-4MI | T925@36h T600@12h | Z1000@12h Z500@24h | 170 | W850@06h W850@12h W850@18h W850@24h | 70 | MI925@12h MI925@24h MI700@12h MI700@24h | 25 | Ben Daoud et al. 2016 |

† or MF925@06h+18h as an alternative